# Revisiting Sparsity Hunting in Federated Learning: Why does Sparsity Consensus Matter?

**Sara Babakniya** *                                    *babakniy@usc.edu*
*Department of Computer Science*
*University of Southern California, USA*
**Souvik Kundu** *                                    *souvikk.kundu@intel.com*
*Intel Labs, San Diego, USA*
**Saurav Prakash**                                    *sauravpr@usc.edu*
*Department of Electrical and Computer Engineering*
*University of Southern California, USA*
**Yue Niu**                                    *yueniu@usc.edu*
*Department of Electrical and Computer Engineering*
*University of Southern California, USA*
**Salman Avestimehr**                                    *avestime@usc.edu*
*Department of Electrical and Computer Engineering*
*University of Southern California, USA*

**Reviewed on OpenReview:** *https://openreview.net/forum?id=iHyhdpsnyi*

## Abstract

Edge devices can benefit remarkably from federated learning due to their distributed nature; however, their limited resource and computing power poses limitations in deployment. A possible solution to this problem is to utilize off-the-shelf sparse learning algorithms at the clients to meet their resource budget. However, such naive deployment in the clients causes significant accuracy degradation, especially for highly resource-constrained clients. In particular, our investigations reveal that the lack of consensus in the sparsity masks among the clients may potentially slow down the convergence of the global model and cause a substantial accuracy drop. With these observations, we present *federated lottery aware sparsity hunting* (FLASH), a unified sparse learning framework for training a sparse sub-model that maintains the performance under ultra-low parameter density while yielding proportional communication benefits. Moreover, given that different clients may have different resource budgets, we present *hetero-FLASH* where clients can take different density budgets based on their device resource limitations instead of supporting only one target parameter density. Experimental analysis on diverse models and datasets shows the superiority of FLASH in closing the gap with an unpruned baseline while yielding up to ∼10.1% improved accuracy with ∼10.26× fewer communication, compared to existing alternatives, at similar hyperparameter settings. Code is available at `https://github.com/SaraBabakN/flash_fl.git`

## 1 Introduction

Federated learning (FL) (McMahan et al., 2017) is a popular form of distributed training, which has gained significant traction due to its ability to allow multiple clients to learn a common global model without sharing their private data. However, clients' heterogeneity and resource limitations pose significant challenges for FL deployment over edge nodes, including mobile and IoT devices. To resolve these issues, various methods have been proposed over the past few years for efficient heterogeneous training (Zhu et al., 2021; Horvath et al., 2021; Diao et al., 2020) or aggregation with faster convergence and reduced communication (Li et al.,

---

*Authors have equal contribution.

2020b; Reddi et al., 2020). However, these methods do not necessarily address the growing concerns of high computation and communication limited edge.

Meanwhile, efficient edge deployment via reducing the memory, compute, and latency costs for deep neural networks in centralized training has also become an active area of research. In particular, recently proposed *sparse learning* strategies (Evci et al., 2020; Kundu et al., 2021; 2022; Mocanu et al., 2018; Dettmers & Zettlemoyer, 2019) effectively train weights and associated binary *sparse masks* such that only a fraction of model parameters (density $d << 1$) are non-zero during training. This enables the potential for reduction in both the training time and compute cost (Qiu et al., 2021; Raihan & Aamodt, 2020), *while yielding accuracy close to that of the unpruned baseline.*

In addition to the aforementioned benefits of sparse learning in centralized settings, its proper deployment in FL can reduce communication costs. In particular, users can train larger models while communicating only a fraction of weights that are non-zero (Fig. 1 (a)).

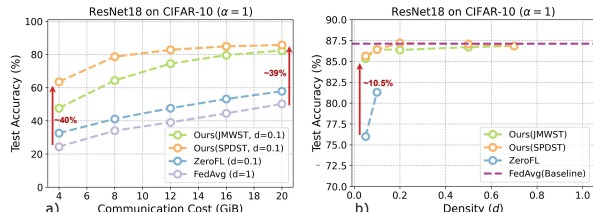

However, the challenges and opportunities of sparse learning in FL are yet to be fully unveiled. Only very recently, few works (Bibikar et al., 2022; Huang et al., 2022) have tried to leverage non-aggressive model compression in FL. Another recent work, ZeroFL (Qiu et al., 2021), has explored deploying sparse learning in FL; however, failed to leverage the advantages of model sparsity to reduce the clients' communication costs. Moreover, as shown in Fig. 1(b), for $d = 0.05$, ZeroFL suffers from a substantial accuracy drop of ~14% w.r.t baseline.

Figure 1: (a) Accuracy vs. Communication. For a given communication threshold, sparse learning in FL can improve performance. (b) Accuracy vs. parameter density in each client. Our approach significantly outperforms the existing alternative (Qiu et al., 2021).

**Contributions.** Our contribution is fourfold. In view of the above, we first present crucial observations in identifying several limitations in integrating sparse learning in federated settings. In particular, we observe that the sparse aggregated model does not converge to a unique sparsity pattern, primarily due to a **lack of consensus** among the clients' masks in different rounds. In contrast, as the model matures in centralized training, the mask also shows a higher convergence trend. We further empirically demonstrate the utility of incorporating layer importance and clients' consensus on the performance.

We then leverage our findings and present *federated lottery aware sparsity hunting* (FLASH), a methodology that can achieve computation and communication efficiency in FL by employing sparse learning.

Furthermore, in real-world scenarios, FL users are more likely to have highly *heterogeneous resource budgets* (Diao et al., 2020). Therefore, instead of limiting everyone by the minimum available resource, we extend our methodology to *hetero*-FLASH, where different clients can participate with different sparsity budgets yet yield a sparse model, leveraging the resource and data of each client while adhering to their own resource limit. Here, to deal with this problem, we propose server-side sub-sampling where the server creates multiple sub-masks of the global model's mask such that all follow the global layer importance.

We conduct experiments on MNIST, FEMNIST, CIFAR-10, CIFAR-100, and TinyImageNet with different models for both IID and non-IID client data partitioning. Our results show that compared to the existing alternative (Qiu et al., 2021), at iso-hyperparameter settings, FLASH can yield up to ~8.9% and ~10.1%, on IID and non-IID data settings, respectively, with reduced communication of up to ~10.2×.

## 2 Related Works

**Model pruning.** In the past few years, a plethora of research focused on model compression via pruning (Frankle & Carbin, 2018; Liu et al., 2021; You et al., 2019; Kundu & Sundaresan, 2021; Niu et al., 2020; Kundu et al., 2023a). Pruning essentially identifies and removes the unimportant weights to yield efficient inference models. To this aim, various methods have been proposed for different purposes. Here, we focus on Sparse Learning, which can potentially reduce the update size and compression errors (Xu et al., 2020) for FL users.

**Sparse learning.** It is a popular form of model pruning that recently has gained significant traction (Evci et al., 2020; Kundu et al., 2020; 2019). In particular, in sparse learning with target density $d$, only $d\%$ of the model parameters remain non-zero during the training ($d << 1.0$ and sparsity is $1.0 - d$).

In our paper, we leverage Dettmers & Zettlemoyer (2019) to find the sparse mask efficiently for each client. Particularly, the training starts with a sparse model and a randomly initiated mask that meets the target parameter density $d$. Since there is no prior knowledge about the importance of weights and layers, we start with a random sparse mask with uniform layer-wise parameter density $d$. The mask gets updated based on a *prune-regrow* policy at the end of each epoch. In each round, first, layers are ranked based on their normalized contribution in the non-zero weights. Then, the lowest $p_r\%$[1] weights from each layer are pruned based on their absolute magnitude. Since this $p_r\%$ pruning happens on top of $d\%$ density, we need to regrow $p_r\%$ from the pruned ones back in the model. Here, each layer gets the number of regrown weights proportional to their normalized contribution, giving layers with higher importance more non-zero weights. This process iteratively repeats each epoch to learn the mask and non-zero weights. Note that there are multiple choices for pruning and regrowing the weights; however, we chose absolute magnitude to save the cost of computing additional values, such as the momentum.

**FL for resource and communication limited edge.** Resource limitation and heterogeneity are among the most known challenges, especially in cross-device federated learning (Kairouz et al., 2021). Existing works have explored the idea of heterogeneous training that allows clients to train on different fractions of full-model based on their compute budget (Horvath et al., 2021; Diao et al., 2020; Mei et al., 2022; Niu et al., 2022). On a parallel track, various optimizations are proposed to accelerate the convergence, thus requiring fewer communication rounds (Han et al., 2020; Gorbunov et al., 2021; Zhang et al., 2013; Li et al., 2019).

To make FL more communication efficient, a few works have leveraged pruning (Li et al., 2020a; Jiang et al., 2022; Li et al., 2021). In LotteryFL (Li et al., 2020a), clients adapt personalized masks that perform well only on their local data. Moreover, the clients must regularly communicate the entire model to the server. Similarly, PruneFL (Jiang et al., 2022) also asks for high communication costs as it demands the participating clients to send the gradient values of the entire model to the server while updating the masks.

Only a few works (Huang et al., 2022; Bibikar et al., 2022; Qiu et al., 2021) tried to benefit from sparse learning in federated settings. In particular, Huang et al. (2022) relied on a randomly initialized sparse mask and recommended keeping it frozen during the training without providing any supporting intuition. FedDST (Bibikar et al., 2022), on the other hand, leveraged RigL (Evci et al., 2020) to perform sparse learning in clients. It relied on a large number of local epochs to avoid gradient noise and focused primarily on highly non-IID data distributions without targeting ultra-low density $d$. More importantly, neither of these works investigated the key differences between sparse learning in centralized and FL. With a similar philosophy as ours, ZeroFL (Qiu et al., 2021) identified a fundamental aspect of sparse learning in FL; the top-k weights of clients are likely to differ for low densities; hence, the aggregated update cannot benefit anyone. They propose increasing the update density to increase the update overlap and improve accuracy. As a result, despite training a model with density $d = 0.1$, clients send updates to the server with $d = 0.3$. Thus, this approach does not benefit from proportional communication efficiency; all clients receive a dense model and send back a $3\times$ denser model, and still, the global model suffers from a significant accuracy drop.

Another contemporary work (Isik et al., 2023) leveraged the idea of learning the sparse mask while keeping the weights fixed to their initialized values (Ramanujan et al., 2020). Thus, clients are only required to send the binary mask updates to the server, reducing the communication by $32\times$. However, due to SGD-based updating of floating point mask for each weight, such methods do not necessarily help the client's local computation. Such an assumption is out of our current scope, as we assume an even stricter constraint where no client can perform dense model updates or storage while yielding significant communication savings.

## 3 Revisiting Sparse Learning: Why Does it Miss the Mark in FL?

In centralized training, applying sparse learning methods has shown benefits in FLOPs reduction during forward operations (Evci et al., 2020), and potential training speed-up of up to $3.3\times$ (Qiu et al., 2021) while

---

[1]Prune rate ($p_r$) indicates the % of weights pruned from each layer of a sparse model.

maintaining high accuracy at low densities ($d \leq 0.1$). However, here, the primary purpose of employing sparse learning is to utilize communication efficiency by reducing update size in FL. Such methods can potentially get better convergence and performance compared to post-training or data-independent methods.

Table 1: FL training settings considered in this work.

| Dataset | Model | Data-partitioning | Rounds ($T$) | Clients ($C_N$) | Clients/Round ($c_r, c_d$) | Optimizer | Aggregation | Local epoch ($E$) | Batch size |
|---|---|---|---|---|---|---|---|---|---|
| MNIST | MNISTNet | | 400 | | | | | | |
| CIFAR-10 | | $LDA$ | 600 | 100 | 10, 10 | SGD | FedAvg | 1 | 32 |
| CIFAR-100 | ResNet18 | | 800 | | | | | | |
| TinyImageNet | | | 600 | | | | | | |
| FEMNIST | Same as (Caldas et al., 2018) | – | 1000 | 3400 | 34, 34 | | | | 16 |

In this section, we conduct an exhaustive analysis to understand how naive deployment of sparse learning in FL works while unveiling key differentiating factors in training dynamics with sparse learning in centralized compared to that of FL. We used a widely adopted FL training setting (Qiu et al., 2021; Diao et al., 2020) (refer to Table 1 for details) on the CIFAR-10 dataset and added sparse learning on the clients. Here, each client separately performs sparse learning (Dettmers & Zettlemoyer, 2019) to train a sparse ResNet18 model and meet a fixed parameter density $d$, starting from a random sparse mask. At the end of each round, clients send their updates to the server, where they get aggregated using FedAvg. We name this *naive sparse training (NST)* due to its plug-and-play nature of deployment of sparse learning in FL. Notably, NST can decrease only communication from clients to the server mainly because these updates usually have different sparsity patterns. Therefore, aggregating them by simply averaging gradients causes the final model to have density $>> d$, translating to a higher downlink communication cost.

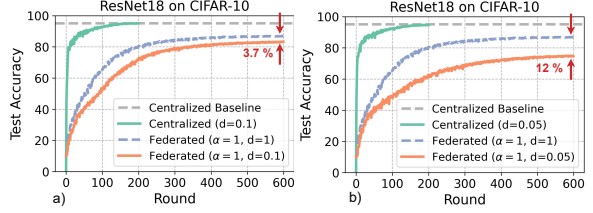

Figure 2: Accuracy vs. round. Comparison between the performance of sparse learning in federated and centralized settings (dense and sparse baselines are shown in dashed and solid lines.).

Figure 3: Heatmap of SM between masks generated in different rounds of (a) centralized and (b) federated learning. Lower SMs indicate more similarities between the masks.

**Observation 1.** *At ultra-low density $d \leq 0.1$, the collaboratively learned FL model significantly sacrifices performance, while the centralized sparse learning yields close to baseline performance.*

Fig. 2 shows that sparse learning methods with high sparsity can perform similarly to the dense model (gray and green lines). However, naively deploying the same method in FL can cause significant performance degradation (purple and orange lines). In particular, the same model can suffer from an accuracy drop of 3.7% and 12% for $d = 0.1$ and $d = 0.05$, respectively.

**Observation 2.** *As the training progresses, the sparse masks in centralized training tend to agree across epochs, showing its convergence, while the server mask in FL does lack consensus across rounds.*

**Definition 1. Sparse mask mismatch.** Let us call the masks generated at rounds $i$ and $j$ as $\mathcal{M}^i$ and $\mathcal{M}^j$. Now, we can define the *sparse mask mismatch*(SM) $\mathtt{sm}^{(i,j)}$ as the Jaccard distance between these two masks where $\mathcal{M}_l^i$ represents the sparse mask tensor for layer $l$.

$$\mathtt{sm}^{(i,j)} = 1 - \frac{(\sum_{l=1}^{L} \mathcal{M}_l^i \cap \mathcal{M}_l^j)}{(\sum_{l=1}^{L} \mathcal{M}_l^i \cup \mathcal{M}_l^j)} \tag{1}$$

Fig. 3 presents the SM for the intermediate sparse masks generated during sparse training in centralized and federated learning. As Fig. 3(a) shows, SM decreases in centralized learning as the training progresses. On the other hand, in Fig. 3(b), the SM values for similar settings (same model, dataset, and density) for the global model in FL remains $> 0.4$ indicating a distinction in the sparse learning between the two settings.

**Observation 3.** *In federated sparse learning with low target density, a lack of consensus at the later (deeper) layer's masks remains more severe than that of the earlier ones.*

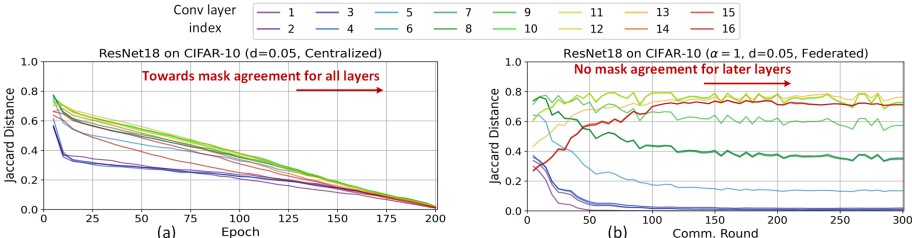

Figure 4: Layer-wise sparse mask mismatch vs. epoch (rounds) for (a) centralized and (b) FL. In FL, the SM for later layers stays high, contrary to centralized, where SM reduces for all layers as the training matures.

Fig. 4 presents the layer-wise SM of the consecutive masks in centralized and federated learning. As shown in Fig. 4(a), by increasing training rounds in centralized learning, for each layer, the distance (measured by SM) between masks decreases, showing a convergence trend. However, in FL (Fig. 4(b)), the later layers' masks change significantly in different rounds with SM value as high as ∼0.8. One possible explanation for this phenomenon is based on layer sensitivity, which we define below.

**Layer sensitivity**. All layers do not carry the same importance towards the final model performance. One proposed metric to measure this difference is layer pruning sensitivity (Ding et al., 2019). After pruning the model using any prospective algorithms, layers that are assigned higher density are the ones that are more sensitive to pruning and play a more important part in the final performance. We can measure the sensitivity of a sparse layer ($l$) via the ratio of its total # of non-zero weights to the total # of weights.

$$\text{sensitivity}^l = \frac{\text{total\# non-zero weights}^l}{\text{total \# weights}^l} \tag{2}$$

It is commonly known that later layers usually have lower sensitivity and a larger number of total parameters, allowing them to have many possible mask options compared to the earlier ones. Therefore, clients are unlikely to reach a consensus and are more likely to come up with different masks in the later layers, causing an increased $SM$. For example, 90% of the parameters in layer 1 and 5% of the parameters in layer 14 are present in the final mask, and as expected, $SM$ for these layers is 0 and ∼0.73, respectively.

To further investigate the impact of high SM and layer sensitivity on a model's accuracy, we performed five different centralized training as described in Table 2. Particularly for the training of row 1, we randomly generate sparse masks with uniform density for all the layers. For rows $2-5$, first, we randomly create each layer's mask by following its pruning sensitivity[2], then decide to freeze (prevent changing) the mask partially or for all layers. More specifically, in row 2, the initialized (pre-defined) mask does not change. For rows $3-5$, at each epoch, a fraction of the mask in the specified layers changes to meet the target SM value. As Table 2 shows that large SM of masks between epochs can degrade the

Table 2: Performance based on the different levels of mask disagreement in centralized.

| Training method | Use sensitivity | Masks change at | Layer SM | Test acc% |
|---|---|---|---|---|
| Pre-defined with frozen mask | N | – | – | 89.72 |
| | Y | – | – | 91.66 |
| Without frozen mask | Y | layer 9-16 | 0.8 | 88.88 |
| | Y | layer 1-16 | 0.5 | 84.62 |
| | Y | layer 1-16 | 0.8 | 82.32 |

accuracy by up to 9.34%, and we can safely conclude that *disagreement of masks across epochs can affect the final performance.* Moreover, the model trained via sparse learning with sensitivity-driven pre-defined masks (row 2) yields better performance than the one trained using a pre-defined mask.

---

[2]We train a separate model with the same architecture and target $d$ to measure the final layer-wise sensitivity.

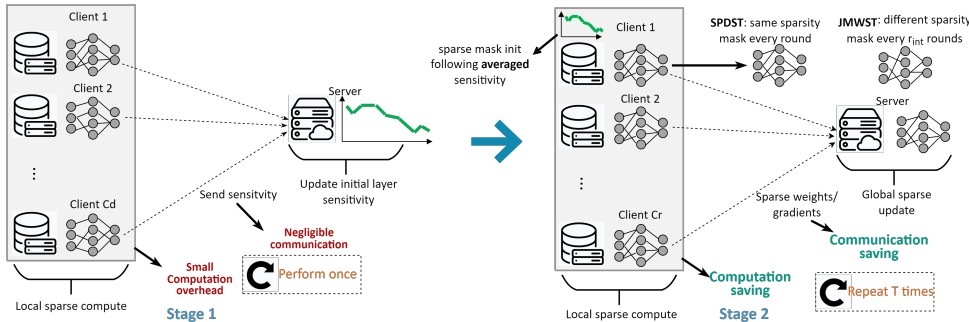

Figure 5: Summary of FLASH. `Stage 1`: $C_d$ clients perform sparse training locally to find layer-wise sensitivities. `Stage 2`: Clients collaboratively train the weights under the server's supervision to follow the layer sensitivities and reach a consensus in yielding sparse masks.

Based on this observation, we hypothesize that such a lack of consensus impacts the model's performance. Changing the mask every epoch means a large chunk of weights to be updated from 0, affecting the trained weights' maturity. We note that this is an observation in a centralized setting. However, we believe it reflects a generic phenomenon irrespective of the centralized or decentralized nature of training.

## 4    FLASH: Methodology

Based on observations in section 3, we hypothesize that deploying sparse learning in clients, though it helps them find a lottery ticket individually, fails in finding a global one. To help FL find the winning lottery; we identify two key characteristics of sparse learning, *pruning sensitivity* and *mask convergence* and present federated lottery aware sparsity hunting (FLASH) methodologies. To adhere to the identified feature, FLASH includes two stages: `Stage 1` that learns layer sensitivity to properly initialize the sparse mask; `Stage 2` that trains weights and masks. Below, we explain how each step works and highlight its importance. Algorithm 1 details the training methods, and Fig. 5 summarizes FLASH and its components.

### 4.1    `Stage 1`: Mask Initialization

This stage aims to identify a good sparse mask the server can provide to the participating clients. However, since in FL, the server does not have access to the training data, the server requires the help of the clients to estimate layer importance. Server first randomly selects a small fraction of clients ($[\mathcal{C}_d]$) and asks them to perform sparse learning (locally) on their private data for a few warm-up epochs ($E_d$) ($L$4-8 in Algorithm 1). It then evaluates each of $c_d$ clients' layer sensitivities via Eq. 2. For each layer $l$, the server estimates average density [3] by averaging the sensitivity of that layer over $c_d$ clients, i.e. $\hat{d}^l = \frac{\sum_{i=1}^{c_d} d_i^l}{c_d}$, where $d_i^l$ is the density at layer $l$ in $i^{th}$ client. As these averaged layer-wise density values may not necessarily satisfy the target density ($d$) criteria, for a model with $K$ parameters, we present the following *density re-calibration* formulation

$$d_c^l = \hat{d}^l.r_f, \text{ where } r_f = \frac{d \times K}{\sum_{l=1}^{L} \hat{d}^l.k^l} \tag{3}$$

$k^l$ is the dense model's parameter size for layer $l$. For each layer $l$ in the model, the server creates a binary sparse mask tensor that is randomly initialized with a fraction of ones $\propto d_c^l$ ($L$9 in Algorithm 1).

### 4.2    `Stage 2`: Sensitivity-Aware Training

Using `Stage 1`, clients can start training with a sensitivity-driven initialized mask. However, this does not guarantee the mask convergence which we have shown (section 3, Obs. 2). Therefore, we propose the following two approaches that can significantly improve mask convergence.

---

[3]which is the same as sensitivity for a layer.

---

**Algorithm 1:** FLASH Training.

---

**Data:** Training rounds $T$, local epochs $E$, clients $[\mathcal{C}_N]$, clients per rounds $c_r$, target density $d$ ,
    sensitivity warm-up epochs $E_d$, density warm-up client $c_d$, $freez = 0$ and Aggregation type $Agr$.

**1** $\Theta^{init} \leftarrow \texttt{initRandomMaskedWeight}(d)$

**2** serverExecute:

**3** # Calculate the layer-wise sensitivity in `stage 1`

**4** Randomly sample $c_d$ clients $[\mathcal{C}_d] \subset [\mathcal{C}_N]$

**5** **for** *each client $c \in [\mathcal{C}_d]$* **in parallel** **do**

**6**   $\Theta_c \leftarrow \texttt{clientExecute}(\Theta^{init}, E_d, 0) \# freez = 0$

**7**   $\mathcal{S}_c \leftarrow \texttt{computeSensitivity}(\Theta_c)$

**8** **end**

**9** $\Theta^0 \leftarrow \texttt{initSensivityDrivenMaskedWeight}([\mathcal{S}_c], d)$

**10** $freez \leftarrow 1$ if SPDST, 0 if JMWST

**11** # Start `Stage 2`

**12** **for** *each round $t \leftarrow 1$* **to** $T$ **do**

**13**   Randomly sample $c_r$ clients $[\mathcal{C}_r] \subset [\mathcal{C}_N]$

**14**   **for** *each client $c \in [\mathcal{C}_r]$* **in parallel** **do**

**15**    $\Theta_c^t \leftarrow \texttt{clientExecute}(\Theta^{t-1}, E, freez)$

**16**   **end**

**17**   $\Theta_S^t \leftarrow \texttt{aggrParamUpdateMask}([\Theta_c^t], Agr)$

**18**   $\Theta^t \leftarrow \texttt{subsampleServerModel}(\Theta_S^t, [\Theta_c^t], d, freez)$

**19** **end**

**20** return $\Theta^T$

**21** clientExecute$(\Theta_{c^0}, E, freez)$ :

**22** **for** *local epoch $i \leftarrow 1$* **to** $E$ **do**

**23**   $\Theta_{c^i} \leftarrow \texttt{doSparseLearning}(\Theta_{c^{i-1}}, freez)$

**24** **end**

**25** return $\Theta_{c^E}$

---

**Approach 1: SPDST.** In this approach, to achieve a convergent mask and low `sm` values, the server pre-defines layer masks at initialization (set $freez = 1$ at $L10$ in Algorithm 1). This way, all the clients agree on the mask and only train the non-zero weights. This guarantees no mask divergence issue ($\texttt{sm}^{(i,j)} = 0$ for all $i, j$). Moreover, as FLASH disentangles the sensitivity evaluation stage from the training, the pre-defined mask in this scenario benefits from the notion of layer sensitivity. We thus aptly name this approach as *sensitivity-driven pre-defined sparse training* (SPDST). Interestingly, earlier research (Bibikar et al., 2022) hinted at poor model performance with pre-defined masks, contrasting ours where we see significantly improved model performance, implying the importance of `stage 1` (as will be elaborated in section 5).

**Approach 2: JMWST.** Here, after `stage 1`, the global mask is not frozen, and clients have the freedom to come up with their mask and change the global one (set $freez = 0$ at $L10$ in Algorithm 1). This way, the model masks and weights are jointly learned during clients' local learning, thus termed as *joint mask weight sparse training* (JMWST). However, as highlighted earlier, clients' naive sparse mask selection at the beginning of each round costs a considerable accuracy drop (section 3 Obs. 1). To avoid this problem and increase mask consensus, JMWST allows the server to subsample a sparse model with density $d$ at round $t + 1$ from the aggregated model at the end of round $t$.

As mentioned in section 3, with target density $d$, the aggregated model has density $d_{server} > d$. To enable efficient sampling of a sparse model while adhering to layer sensitivity, we leverage the density re-calibration strategy (Eq. 3) by taking the $t^{th}$ round's clients' average sensitivity into consideration ($L18$ in Algorithm 1). The server performs magnitude pruning to retain the top-$d_c^l$ fraction of parameters for $l^{th}$ layer and sends the pruned model to clients at round $t + 1$. Intuitively, the server's sampling of non-zero weights reduces the chances of wasted updates and accelerates mask convergence due to alignment with the layers' pruning

sensitivity. Then, the next round's clients can perform sparse learning locally, yield another set of sparse models, send them to the server, and so on. By default, the server evaluates masks every round $r_{int} = 1$. However, it can increase the update interval, and clients update the mask after a specific $r_{int} > 1$ round.

**SPDST vs. JMWST.** The update aggregation and communication are more straightforward in SPDST, as the position of the zero and non-zero parameters are fixed, and the global model's density remains at $d$ both at the server and the clients. In terms of mask convergent, SM is always 0 for SPDST because the mask is pre-defined and does not change. In contrast, JMWST gives more freedom to the client, and they can participate in mask training. Additionally, we observed that JMWST has a lower SM than NST, and the mask it more convergent. For example, for the CIFAR-10 dataset after 300 rounds, the SM value for JMWST is $\sim 85\%$ lower than that of NST.

**Extension to support heterogeneous density.** The current framework assumes that all the clients train with the same target density $d$. However, users may have different resource limitations, and some may contribute more resources to the training based on their budget. We now present hetero-FLASH to support this resource-heterogeneity of users and adhere to their respective density budgets.

Let us assume there are $N$ available support densities $d_{set} = [d_1, .., d_N]$, where $d_i < d_{i+1}$. Now, for hetero-SPDST and hetero-JMWST, we perform the same `Stage 1` as explained before to create the masks for the clients with the highest density $d_N$. For any other density $d_i$, we sample a sparse mask from that with density $d_{i+1}$. Note, while creating the mask from $d_{i+1}$ to $d_i$, we follow the layer-wise density re-calibration approach (Eq. 3). Similar to the original SPDST, after `Stage 1`, the server freezes all the $N$ pre-defined mask for the heterogeneous case, and for the rest of the training, clients can use the mask associated with their budget. For hetero-JMWST, at the beginning of each round, the server performs magnitude pruning to yield $N$ sub-models meeting $N$ different density levels, contrasting to the creation of one model in JMWST. Participating clients of different densities use the corresponding sub-models to start sparse learning locally.

In hetero-FLASH, clients do not contribute to weights equally. Considering all the parameters from clients with sparse masks, specifically, the pruned parameters may hurt the updates that are not zero on that same position. Therefore, instead of FedAvg., the server performs aggregation we call *Weighted Fed Averaging* (WFA). In particular, with similar inspiration as Diao et al. (2020), instead of averaging over # of participating clients, we average each non-zero update by their total non-zero occurrences over the participating clients in a round. We have provided the algorithm for hetero-FLASH in the Appendix.

## 5 Experiments

**Datasets and models.** We evaluated the performance of FLASH on MNIST (LeCun & Cortes, 2010) on MNISTNet (McMahan et al., 2017), FEMNIST on model described in (Caldas et al., 2018) and CIFAR-10, CIFAR-100 (Krizhevsky et al., 2009) and TinyImageNet (Pouransari & Ghili, 2014) on ResNet18 (Further details in the Appendix). For data partitioning of MNIST, CIFAR-10, CIFAR-100, and TinyImageNet, we use Latent Dirichlet Allocation (LDA)(Reddi et al., 2020) with three different $\alpha$ ($\alpha = 1000, 1, 0.1$). In this allocation, decreasing the value of the *alpha* increases the data heterogeneity among the clients. FEMNIST is built from handwriting 3400 different users (Han et al., 2020), making it inherently non-IID.

**Training hyperparameters.** We use starting learning rate ($\eta_{init}$) as 0.1 which exponentially decayed to 0.001 ($\eta_{end}$). Specifically, learning rate for participants at round t is $\eta_t = \eta_{init}(\exp(\frac{t}{T}\log(\frac{\eta_{init}}{\eta_{end}})))$. In all the sparse learning experiments, the pruning rate is $0.25$[4]. Other training hyperparameters can be found in Table 1. Furthermore, all the results are averaged over three different seeds.

### 5.1 Experimental Results with FLASH

To understand the importance of `stage 1` in FLASH methodology, we identify a baseline training with uniform layer sensitivity-driven *pre-defined sparse training* (PDST) in FL. Table 3 details the performance of FLASH at different levels of $d$. In particular, as we can see in Table 3 columns 5 and 6, the performance of the model for both NST and PDST drops at ultra-low parameter density $d = 0.05, 0.01$. For example,

---

[4]Prune rate controls the fraction of non-zero weights participating in the prune-regrow during sparse learning.

Table 3: Results for different datasets with FLASH (SPDST, and JMWST) and its comparison with NST and PDST for different densities ($d \in \{5\%, 10\%\}$ for all datasets and extreme sparse, $d = 1\%$ for CIFAR-10, CIFAR-100 and TinyImageNet) and data distributions ($\alpha \in \{0.1, 1, 1000\}$).

| Dataset | Data Distribution | Density (d) | Baseline Acc % | NST Acc % | PDST Acc % | SPDST Acc % | JMWST($r_{int}=1$) Acc % | JMWST($r_{int}=5$) Acc % |
|---|---|---|---|---|---|---|---|---|
| MNIST | IID ($\alpha = 1000$) | 1.0 | $98.79 \pm 0.06$ | – | – | – | – | – |
| | | 0.1 | – | $97.57 \pm 0.11$ | $97.09 \pm 0.18$ | $\mathbf{98.21 \pm 0.06}$ | $97.95 \pm 0.16$ | $98.09 \pm 0.16$ |
| | | 0.05 | – | $95.19 \pm 0.56$ | $94.8 \pm 1.04$ | $\mathbf{97.46 \pm 0.14}$ | $97.24 \pm 0.21$ | $97.37 \pm 0.23$ |
| | non-IID ($\alpha = 1.0$) | 1.0 | $98.76 \pm 0.06$ | – | – | – | – | – |
| | | 0.1 | – | $97.36 \pm 0.19$ | $96.82 \pm 0.25$ | $97.96 \pm 0.13$ | $97.72 \pm 0.12$ | $\mathbf{98.11 \pm 0.12}$ |
| | | 0.05 | – | $95.75 \pm 0.31$ | $95.34 \pm 0.77$ | $97.3 \pm 0.26$ | $97.38 \pm 0.11$ | $\mathbf{97.59 \pm 0.07}$ |
| | non-IID ($\alpha = 0.1$) | 1.0 | $98.45 \pm 0.17$ | – | – | – | – | – |
| | | 0.1 | – | $96.19 \pm 0.22$ | $94.41 \pm 1.23$ | $\mathbf{97.22 \pm 0.43}$ | $96.53 \pm 0.19$ | $96.7 \pm 0.14$ |
| | | 0.05 | – | $91.66 \pm 1.74$ | $91.06 \pm 1.1$ | $95.7 \pm 0.37$ | $95.83 \pm 0.84$ | $\mathbf{95.91 \pm 0.64}$ |
| CIFAR-10 | IID ($\alpha = 1000$) | 1.0 | $88.56 \pm 0.06$ | – | – | – | – | – |
| | | 0.1 | – | $84.89 \pm 0.26$ | $86.72 \pm 0.09$ | $\mathbf{88 \pm 0.28}$ | $87.62 \pm 0.35$ | $87.86 \pm 0.13$ |
| | | 0.05 | – | $77.48 \pm 0.54$ | $84.38 \pm 0.12$ | $86.99 \pm 0.14$ | $86.87 \pm 0.08$ | $\mathbf{87.18 \pm 0.09}$ |
| | | 0.01 | – | $52.70 \pm 1.17$ | $70.17 \pm 0.70$ | $82.35 \pm 0.14$ | $83.59 \pm 0.38$ | $\mathbf{83.85 \pm 0.26}$ |
| | non-IID ($\alpha = 1.0$) | 1.0 | $87.13 \pm 0.18$ | – | – | – | – | – |
| | | 0.1 | – | $83.46 \pm 0.19$ | $85.07 \pm 0.24$ | $\mathbf{86.42 \pm 0.49}$ | $\mathbf{86.45 \pm 0.31}$ | $86.36 \pm 0.13$ |
| | | 0.05 | – | $75.1 \pm 0.76$ | $83.33 \pm 0.14$ | $85.64 \pm 0.58$ | $85.34 \pm 0.27$ | $\mathbf{85.9 \pm 0.24}$ |
| | | 0.01 | – | $50.71 \pm 0.99$ | $69.44 \pm 0.63$ | $81.01 \pm 0.50$ | $\mathbf{82.38 \pm 0.18}$ | $82.31 \pm 0.12$ |
| | non-IID ($\alpha = 0.1$) | 1.0 | $77.64 \pm 0.49$ | – | – | – | – | – |
| | | 0.1 | – | $71.18 \pm 1.23$ | $74.82 \pm 0.72$ | $\mathbf{76.74 \pm 1.46}$ | $74.74 \pm 1.07$ | $75.47 \pm 1.18$ |
| | | 0.05 | – | $61.29 \pm 2.76$ | $72.32 \pm 1.05$ | $75.47 \pm 2.31$ | $73.9 \pm 1.45$ | $\mathbf{75.49 \pm 0.9}$ |
| | | 0.01 | – | $42.66 \pm 0.50$ | $59.30 \pm 0.36$ | $70.61 \pm 1.82$ | $68.89 \pm 0.62$ | $\mathbf{71.21 \pm 1.98}$ |
| CIFAR-100 | IID ($\alpha = 1000$) | 1.0 | $65.38 \pm 0.27$ | – | – | – | – | – |
| | | 0.1 | – | $53.81 \pm 0.92$ | $61.16 \pm 0.51$ | $62.35 \pm 0.40$ | $\mathbf{62.88 \pm 0.26}$ | $62.69 \pm 0.24$ |
| | | 0.05 | – | $42.08 \pm 0.48$ | $56.67 \pm 0.25$ | $60.32 \pm 0.27$ | $59.59 \pm 0.19$ | $\mathbf{60.29 \pm 0.16}$ |
| | | 0.01 | – | $22.64 \pm 0.75$ | $38.99 \pm 1.16$ | $49.67 \pm 0.49$ | $51.53 \pm 0.76$ | $\mathbf{51.81 \pm 0.13}$ |
| | non-IID ($\alpha = 1.0$) | 1.0 | $65.17 \pm 0.27$ | – | – | – | – | – |
| | | 0.1 | – | $53.36 \pm 0.51$ | $60.87 \pm 0.40$ | $\mathbf{62.13 \pm 0.26}$ | $61.59 \pm 0.07$ | $61.66 \pm 0.11$ |
| | | 0.05 | – | $42.48 \pm 0.39$ | $56.57 \pm 0.28$ | $59.57 \pm 0.35$ | $59.27 \pm 0.62$ | $\mathbf{59.85 \pm 0.35}$ |
| | | 0.01 | – | $23.39 \pm 0.37$ | $38.99 \pm 0.34$ | $49.05 \pm 0.40$ | $50.60 \pm 0.10$ | $\mathbf{51.61 \pm 0.66}$ |
| | non-IID ($\alpha = 0.1$) | 1.0 | $59.12 \pm 0.63$ | – | – | – | – | – |
| | | 0.1 | – | $49.04 \pm 0.57$ | $55.06 \pm 0.26$ | $\mathbf{56.79 \pm 0.33}$ | $54.74 \pm 0.68$ | $55.54 \pm 0.71$ |
| | | 0.05 | – | $37.33 \pm 0.39$ | $51.68 \pm 0.32$ | $\mathbf{54.34 \pm 0.17}$ | $52.67 \pm 0.97$ | $53.47 \pm 0.49$ |
| | | 0.01 | – | $19.21 \pm 0.19$ | $35.59 \pm 0.26$ | $45.10 \pm 0.64$ | $45.31 \pm 0.57$ | $\mathbf{46.16 \pm 0.76}$ |
| TinyImageNet | IID ($\alpha = 1000$) | 1.0 | $55.36 \pm 0.25$ | – | – | – | – | – |
| | | 0.1 | – | $44.63 \pm 0.17$ | $51.95 \pm 0.11$ | $\mathbf{53.18 \pm 0.41}$ | $52.05 \pm 0.09$ | $52.51 \pm 0.35$ |
| | | 0.05 | – | $38.39 \pm 0.14$ | $48.61 \pm 0.25$ | $\mathbf{51.31 \pm 0.41}$ | $50.37 \pm 0.48$ | $50.53 \pm 0.54$ |
| | | 0.01 | – | $20.50 \pm 0.43$ | $37.85 \pm 0.07$ | $43.66 \pm 0.35$ | $43.07 \pm 0.80$ | $\mathbf{44.41 \pm 0.17}$ |
| | non-IID ($\alpha = 1.0$) | 1.0 | $54.76 \pm 0.35$ | – | – | – | – | – |
| | | 0.1 | – | $44.48 \pm 0.16$ | $50.50 \pm 0.06$ | $\mathbf{52.75 \pm 0.18}$ | $51.22 \pm 0.30$ | $51.81 \pm 0.08$ |
| | | 0.05 | – | $38.03 \pm 0.27$ | $47.52 \pm 0.29$ | $\mathbf{51.07 \pm 0.23}$ | $49.48 \pm 0.23$ | $49.76 \pm 0.48$ |
| | | 0.01 | – | $20.88 \pm 0.14$ | $37.49 \pm 0.52$ | $42.91 \pm 0.24$ | $43.04 \pm 0.12$ | $\mathbf{43.15 \pm 0.25}$ |
| | non-IID ($\alpha = 0.1$) | 1.0 | $48.12 \pm 0.16$ | – | – | – | – | – |
| | | 0.1 | – | $38.92 \pm 0.23$ | $42.64 \pm 0.40$ | $\mathbf{46.25 \pm 0.07}$ | $45.18 \pm 0.38$ | $45.21 \pm 0.18$ |
| | | 0.05 | – | $32.81 \pm 0.66$ | $40.69 \pm 0.43$ | $\mathbf{44.56 \pm 0.10}$ | $43.31 \pm 0.19$ | $44.29 \pm 0.16$ |
| | | 0.01 | – | $18.01 \pm 0.13$ | $33.79 \pm 0.15$ | $37.80 \pm 0.06$ | $37.32 \pm 0.51$ | $\mathbf{38.32 \pm 0.19}$ |
| FEMNIST | non-IID | 1.0 | $84.68 \pm 0.20$ | – | – | – | – | – |
| | | 0.1 | – | $76.92 \pm 0.42$ | $76.01 \pm 1.26$ | $82.70 \pm 0.26$ | $83.02 \pm 0.21$ | $\mathbf{83.4 \pm 0.26}$ |
| | | 0.05 | – | $37.33 \pm 0.39$ | $51.68 \pm 0.32$ | $\mathbf{54.34 \pm 0.17}$ | $52.67 \pm 0.97$ | $53.47 \pm 0.49$ |

on CIFAR-10 ($\alpha = 0.1$), models from NST and PDST sacrifice accuracy of 16.35% and 5.32%, respectively. However, at comparatively higher density ($d = 0.1$), both can yield models with a lower accuracy difference from the baseline by around 6.46% and 2.82%. SPDST, on the other hand, can maintain **close to the baseline accuracy** at even ultra-low density for all data partitions and, interestingly, even outperform JMWST ($r_{int} = 1$) for the majority of the cases. *These results highlight the efficacy of both sensitivity-driven sparse learning (as SPDST > PDST) and early mask convergence (as SPDST $\approx$ JMWST) in FL settings. Importantly, for increased $r_{int}$ in JMWST, we observe a consistent improvement in accuracy.* The inferior accuracy at $r_{int} = 1$ can be attributed to the mask divergence caused by frequent noisy gradient updates. We thus believe efficient hyperparameter search including $r_{int}$ is essential for the sparse FL model's improved performance. Also note that JMWST requires additional communication of non-zero weight indices for rounds that masks are updating, contrasting SPDST, with clients not needing to send the mask at all,

*allowing us to yield proportional communication saving as the model density.* Fig. 6 shows the accuracy vs. comm. round for the two proposed methods on different data distributions.

**Extreme sparse learning.** For CIFAR-10, CIFAR-100, and TinyImageNet, which are trained on a comparably more complex model (ResNet18), we also explore the extremely sparse training scenario where clients train only 1% of the weights ($100\times$ saving in communication). This experiment shows the power and importance of both stages in FLASH in achieving a good performance. In contrast, the performance gap between the dense model and NST or PDST remarkably increases.

**Comparison with ZeroFL.** Despite leveraging a form of sparse learning (Raihan & Aamodt, 2020), ZeroFL requires a higher up-link/down-link budget than the target density $d$ to achieve better performance. However, FLASH reaches considerably better accuracy while being substantially more efficient in communication compared to ZeroFL. Especially for SPDST with a frozen mask, the server and clients only need to communicate updates with density $d$ and enjoy bidirectional savings. Note that every sparse matrix representation requires the value and location of non-zero parameters, but SPDST can eliminate sending the latter as it does not change. Finally, we evaluate the communication saving as the ratio of the dense model size and corresponding sparse model size represented in compressed sparse row (CSR) format (Tinney & Walker, 1967). As depicted in Table 4[5], FLASH can yield an accuracy improvement of up to 10.1% at a reduced communication cost of up to $10.26\times$ (at up-link when both send sparse models).

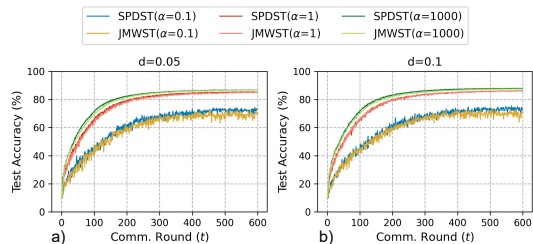

Figure 6: Test accuracy vs. round for CIFAR-10 with (a) $d = 0.05$ (b) $d = 0.1$.

Table 4: Comparison with ZeroFL on various performance metrics. (ZeroFL values are the results with the higher accuracy and taken from the original manuscript.)

| Dataset | Data Distribution | Method | Density | Acc% | Down-link Savings | Up-link Savings |
|---|---|---|---|---|---|---|
| CIFAR-10 | IID | ZeroFL (Qiu et al., 2021) | 0.1 | $82.71 \pm 0.37$ | $1\times$ | $1.6\times$ |
| | | SPDST (ours) | 0.1 | $\mathbf{88 \pm 0.28}$ | $\mathbf{9.8\times}$ | $\mathbf{9.8\times}$ |
| | | ZeroFL (Qiu et al., 2021) | 0.05 | $78.22 \pm 0.35$ | $1\times$ | $1.9\times$ |
| | | SPDST (ours) | 0.05 | $\mathbf{86.99 \pm 0.14}$ | $\mathbf{19.5\times}$ | $\mathbf{19.5\times}$ |
| | non-IID ($\alpha = 1.0$) | ZeroFL (Qiu et al., 2021) | 0.1 | $81.04 \pm 0.28$ | $1\times$ | $1.6\times$ |
| | | SPDST (ours) | 0.1 | $\mathbf{86.42 \pm 0.49}$ | $\mathbf{9.8\times}$ | $\mathbf{9.8\times}$ |
| | | ZeroFL (Qiu et al., 2021) | 0.05 | $75.54 \pm 1.15$ | $1\times$ | $1.9\times$ |
| | | SPDST (ours) | 0.05 | $\mathbf{85.64 \pm 0.58}$ | $\mathbf{19.5\times}$ | $\mathbf{19.5\times}$ |
| FEMNIST | non-IID | ZeroFL (Qiu et al., 2021) | 0.05 | $77.16 \pm 2.07$ | $1\times$ | $\mathbf{17.7\times}$ |
| | | SPDST (ours) | 0.05 | $\mathbf{81.18 \pm 0.36}$ | $\mathbf{14.6\times}$ | $14.6\times$ |

## 5.2 Experimental Results with Hetero-FLASH

Table 5 shows the performance of hetero-FLASH for the scenario where the clients can have three possible density budgets defined by the $d_{set}$ with maximum clients' density $d_{max} = 0.2$. Also, we assume 40%, 30%, and 30% of total clients can train models with a density equal to 0.2, 0.15, and 0.1, respectively. The server samples 10% from each set for every round with the corresponding target density. Similar to the trend in FLASH, hetero-SPDST outperforms the hetero-JMWST ($r_{int} = 1$), and increasing mask update interval ($r_{int}$) helps improve its performance (roughly $> 3\%$).

## 5.3 Quantitative Analysis on FLASH's Design Parameters

**Impact of initial sensitivity warm-up of participating clients.** For a realistic scenario, `Stage 1` needs to be efficient and practically feasible. To be more precise, we cannot expect participating clients to train

---

[5]We understand for FEMNIST, ZeroFL reported significantly higher up-link saving; however, to the best of our understanding, it should be similar to their report on other datasets, i.e. $\sim 1.9\times$.

Table 5: Performance of hetero-FLASH on various datasets where each client can have a density from the set $d_{set} \in [0.1, 0.15, 0.2]$ based on their budget, Note that the density of the final model depends on $d_{max}$.

| Dataset | Data Distribution | Max $d_{set}$ | Hetero-SPDST Acc % | Hetero-JMWST ($r_{int} = 1$) Acc % | Hetero-JMWST ($r_{int} = 5$) Acc % |
|---------|-------------------|-------|--------------------|-------------------------------------|-------------------------------------|
| MNIST | IID ($\alpha = 1000$) | 0.2 | $\mathbf{98.29 \pm 0.05}$ | $97.44 \pm 0.23$ | $97.83 \pm 0.10$ |
| | non-IID ($\alpha = 1.0$) | | $\mathbf{98.29 \pm 0.09}$ | $97.47 \pm 0.22$ | $97.80 \pm 0.23$ |
| | non-IID ($\alpha = 0.1$) | | $\mathbf{97.63 \pm 0.22}$ | $96.11 \pm 0.75$ | $96.25 \pm 0.86$ |
| CIFAR-10 | IID ($\alpha = 1000$) | 0.2 | $87.19 \pm 0.26$ | $86.37 \pm 0.2$ | $\mathbf{87.39 \pm 0.15}$ |
| | non-IID ($\alpha = 1.0$) | | $86.16 \pm 0.04$ | $84.67 \pm 0.06$ | $\mathbf{86.19 \pm 0.24}$ |
| | non-IID ($\alpha = 0.1$) | | $\mathbf{75.23 \pm 1.26}$ | $71.3 \pm 2.75$ | $74.34 \pm 0.85$ |
| CIFAR-100 | IID ($\alpha = 1000$) | 0.2 | $63.4 \pm 0.2$ | $60.5 \pm 0.8$ | $\mathbf{62.91 \pm 0.04}$ |
| | non-IID ($\alpha = 1.0$) | | $62.1 \pm 0.2$ | $59.6 \pm 0.2$ | $\mathbf{62.68 \pm 0.28}$ |
| | non-IID ($\alpha = 0.1$) | | $56.4 \pm 0.4$ | $51.5 \pm 0.9$ | $\mathbf{55.98 \pm 0.19}$ |
| TinyImageNet | IID ($\alpha = 1000$) | 0.2 | $51.28 \pm 0.11$ | $49.73 \pm 0.20$ | $\mathbf{52.07 \pm 0.22}$ |
| | non-IID ($\alpha = 1$) | | $50.94 \pm 0.17$ | $49.32 \pm 0.16$ | $\mathbf{51.61 \pm 0.17}$ |
| | non-IID ($\alpha = 0.1$) | | $44.48 \pm 0.40$ | $42.72 \pm 0.61$ | $\mathbf{44.65 \pm 0.45}$ |
| FEMNIST | non-IID | 0.2 | $\mathbf{82.58 \pm 0.24}$ | $82.2 \pm 0.42$ | $82.5 \pm 0.55$ |

their models for a long time in order to get an accurate estimation of layer sensitivity. Also, it is unlikely to access many clients in a single time slot, especially in cross-device federated learning. Therefore, we designed six different scenarios to understand the impact of these parameters on the final model's performance. In particular, we used two different values of participating clients ($[10, 20]$) and three local epoch choices ($[10, 20, 40]$). As shown in Fig. 7(a), the yielded pruning sensitivity follows a similar trend. Moreover, SPDST with a mask chosen from any of these sensitivity lists finally yields FL models with similar performances (Fig. 7(b)), clearly demonstrating the robustness of our warm-up based sensitivity evaluation `stage 1`.

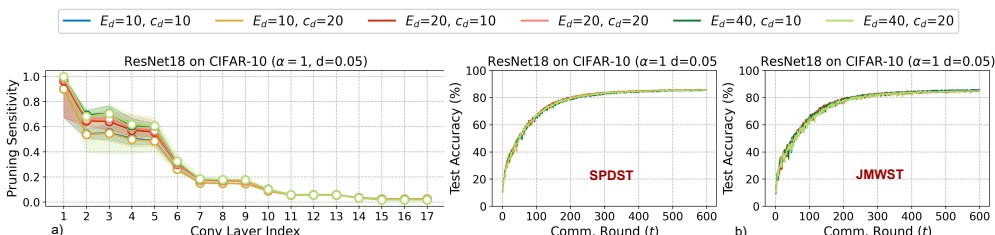

Figure 7: (a) Layer sensitivity evaluated at the end of sensitivity warm-up stage (`Stage 1`) for different client participation ($c_d$) and their local epochs ($E_d$), (b) Comparison of global model performance with the initialized sparse mask based on different sensitivity evaluated from (a).

**Overheads of `stage 1`.** `Stage 1` uses one round with $E_d$ local epochs (here, $E_d = 10$) per client. A normal FL stage in our settings trains the clients for T rounds, 1 epoch per client/round. Hence, this stage increases the time by a factor of $(\frac{E_d}{T} + 1)$. Usually, $E_d << T$, making the pre-training overhead negligible.

The communication overhead of `Stage 1` is also negligible compared to that in each round for the `Stage 2` FL training. Each participant only needs to send $L$ values for an $L$-layer model. So, $c_d$ clients will have a total communication overhead of $(L \times c_d \times 32)$ bits, assuming 32-bit number representation.

**Convergence versus communication costs.** Fig. 8 shows the performance of FL models when the clients have a limited communication budget. In particular, PDST and SPDST can significantly

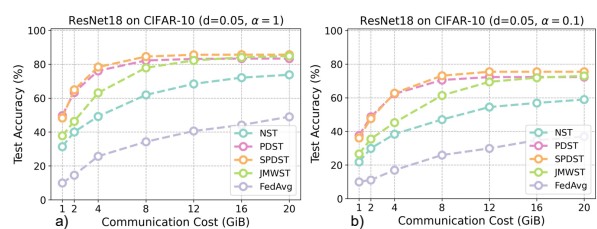

Figure 8: Performance vs. up-link bandwidth for (a) $\alpha = 1.0$ and (b) $\alpha = 0.1$.

outperform other approaches at low communication budgets (even FedAvg). This can be attributed to

their substantially smaller model sizes, helping them to communicate more rounds than others on a limited bandwidth scenario.

**Importance weighted aggregation in hetero-FLASH.** Earlier literature (Diao et al., 2020) suggested weighted averaging in the aggregation of models with different sizes, which we also investigate here. In particular, we performed experiments on CIFAR-10 ($\alpha = 1.0$), both with and without WFA. First, we observe that WFA degrades model performance in JMWST compared to FedAvg (Fig. 9 (a)). On the contrary, the use of WFA improves accuracy for hetero-FLASH (Fig. 9 (b)). The inferior performance of WFA in FLASH may hint at the fact that if a parameter is non-zero only for fewer clients, as compared to other non-zero weights, giving it equal weight as the others in the aggregation nullifies its lower importance, that may be necessary to preserve for mask convergence. On the other hand, having WFA in hetero FLASH is necessary, as the less frequent non-zero occurrence of a parameter can be a result of the presence of fewer high-parameter density clients in a round.

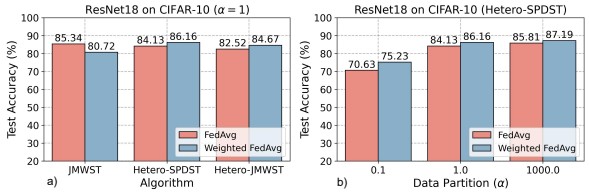

Figure 9: Performance comparison between FedAvg and weighted FedAvg for different (a) algorithms (b) data distributions ($\alpha$).

Figure 10: Performance of models trained with different mask initialization in `stage 1` for target parameter density for (a) $d = 0.05$ (b) $d = 0.1$

**Comparison with ERK+ initialization.** We now compare our SPDST mask initialization with that of parameter density distribution evaluated via ERK+ (Huang et al., 2022; Evci et al., 2020). In contrast with uniform density, the ERK+ scheme keeps more weights for the layers with fewer parameters. To this aim, we use `Stage 1` in SPDST, ERK+, or uniform (PDST) as the initial mask for `stage 2` and keep the mask frozen for the rest of the training. As shown in Fig. 10, the mask initialization using `stage 1` for SPDST consistently provides superior results over the other two. We hypothesize this is rooted in the data-driven layer sensitivity evaluation scheme of SPDST, particularly at the earlier layers, allowing it to retain more information at these layers.

**Ablation on `Stage 1`.** Our proposed method consists of two stages: sensitivity evaluation (`Stage 1`) and training in federated settings (`Stage 2`). In Table 6, we present ablation with and without `Stage 1` for SPDST and JMWST. It is notable that SPDST without `Stage 1` is PDST.

Table 6: Impact of `Stage 1` on final performance on CIFAR-10 dataset with target $d = 0.05$

| Data Distribution | Method | without `Stage 1` | with `Stage 1` |
|---|---|---|---|
| IID($\alpha = 1000$) | SPDST | $84.38 \pm 0.12$ | $\mathbf{86.99 \pm 0.14}$ |
| non-IID($\alpha = 0.1$) | SPDST | $72.32 \pm 1.05$ | $\mathbf{75.47 \pm 2.31}$ |
| IID($\alpha = 1000$) | JMWST | $86.93 \pm 0.1$ | $\mathbf{87.18 \pm 0.09}$ |
| non-IID($\alpha = 0.1$) | JMWST | $74.7 \pm 1.7$ | $\mathbf{75.49 \pm 0.9}$ |

# 6 Conclusion

This paper presented methodologies to yield sparse server models with insignificant accuracy drops compared to the unpruned counterparts. In particular, we demonstrated two efficient sparse learning solutions specifically tailored for FL, enabling better computation and communication benefits over existing sparse learning alternatives. Additionally, we presented the effectiveness of the proposed algorithms for clients with different parameter budgets, allowing deployment for resource-limited edge devices having heterogeneous resource support. The future research direction of this work includes a theoretical understanding of our observations and further empirical demonstrations of the newer class of foundation models.

## 7 Acknowledgement

This material is based upon work supported by ONR grant N00014-23-1-2191, ARO grant W911NF-22-1-0165, Defense Advanced Research Projects Agency (DARPA) under Contract No. FASTNICS HR001120C0088 and HR001120C0160, and gifts from Intel and Qualcomm. The views, opinions, and/or findings expressed are those of the author(s) and should not be interpreted as representing the official views or policies of the Department of Defense or the U.S. Government.

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

# A  Appendix

## A.1  Hetero-FLASH Algorithm

Algorithm 2 details the training algorithm in hetero-FLASH. Note that and `aggrParamUpdateMask` and `subSampleServerModel` are the two functions that play a key role in supporting heterogeneity in sparsity ratios for different clients. The details of these two functions are elaborated in Algorithm 3 and Algorithm 4, respectively. We plan to open-source our code upon acceptance of the paper.

---

**Algorithm 2:** Hetero-FLASH Training.

**Data:** Training rounds $T$, local epochs $E$, client set $[[\mathcal{C}_{N_1}], ..., [\mathcal{C}_{N_M}]]$, clients per rounds $c_r$, target density set $d_{set} = [d_1, ..., d_M]$, sensitivity warm-up epochs $E_d$, density warm-up client count $c_d$, initial value of freeze masks $freez = 0$, training algorithm $A$ and aggregation type $Agr$.

**1**  $\Theta^{init} \leftarrow \texttt{initRandomMaskedWeight}(d_M)$
**2**  serverExecute:
**3**  Randomly sample $c_d$ clients $[\mathcal{C}_d] \subset [\mathcal{C}_{N_M}]$
**4**  **for**  *each client $c \in [\mathcal{C}_d]$ **in parallel**  **do**
**5**  $\quad$ $\Theta_c \leftarrow \texttt{clientExecute}(\Theta^{init}, E_d, 0)$
**6**  $\quad$ $\mathcal{S}_c \leftarrow \texttt{computeSensitivity}(\Theta_c)$
**7**  **end**
**8**  $\Theta^0 \leftarrow \texttt{initSensivityDrivenMaskedWeight}([\mathcal{S}_c], d_{set})$
**9**  $freez \leftarrow \texttt{freezeMask}(A)$
**10**  **for** *each round $t \leftarrow 1$ **to** $T$ **do**
**11**  $\quad$ Randomly sample $c_r$ clients $[\mathcal{C}_r] \subset [\mathcal{C}_N]$
**12**  $\quad$ **for** *each client $c \in [\mathcal{C}_r]$ **in parallel**  **do**
**13**  $\quad\quad$ $\Theta_c^t \leftarrow \texttt{clientExecute}(\Theta^{t-1}, E, freez)$
**14**  $\quad$ **end**
**15**  $\quad$ $\Theta_S^t \leftarrow$ `aggrParamUpdateMask` $([\Theta_c^t], Agr)$
**16**  $\quad$ $\Theta^t \leftarrow$ `subSampleServerModel` $(\Theta_S^t, [\Theta_c^t], d_{set}, freez)$
**17**  **end**
**18**  clientExecute$(\Theta_{c^0}, E, freez)$ :
**19**  **for** *local epoch $i \leftarrow 1$ **to** $E$ **do**
**20**  $\quad$ $\Theta_{c^i} \leftarrow \texttt{doSparseLearning}(\Theta_{c^{i-1}}, freez)$
**21**  **end**
**22**  return $\Theta_{cE}$

---

**Algorithm 3:** `aggrParamUpdateMask`

**Data:** Round $t$, aggregation type $Agr$ [`fedAvg`, `weightedFedAvg`], clients updates $[\Theta^t] = [\Theta_{c_1}, ..., \Theta_{c_r}]$, client data size $[ds_{c_1}, ..., ds_{c_r}]$

**1**  **if** *Agr is  fedAvg* **then**
**2**  $\quad$ $\Theta_S^t \leftarrow \frac{1}{\Sigma_{c_i=1}^{c_r} ds_{c_i}} \Sigma_{c_i=1}^{c_r} ds_{c_i} \cdot \Theta_{c_i}^t$
**3**  **else**
**4**  $\quad$ //For hetero-FLASH
**5**  $\quad$ $\mathcal{W}^t \leftarrow \texttt{initWeightFactor}()$
**6**  $\quad$ **for** *each update $\Theta_{c_i} \in [\Theta^t]$ **do**
**7**  $\quad\quad$ $\mathcal{W}_{c_i}^t \leftarrow ds_{c_i} \times \texttt{retrieveMask}(\Theta_{c_i})$
**8**  $\quad\quad$ $\mathcal{W}^t \leftarrow \mathcal{W}^t + \mathcal{W}_{c_i}^t$
**9**  $\quad$ **end**
**10**  $\quad$ //`safeDivide`(a,b): gives zero anywhere the b is equal to zero
**11**  $\quad$ $\Theta_S^t \leftarrow \Sigma_{c_i=1}^{c_r}[\texttt{safeDivide}(\mathcal{W}_{c_i}^t, \mathcal{W}^t) \cdot \Theta_{c_i}^t]$
**12**  **end**

---

---

**Algorithm 4:** `subsampleServerModel`

---

**Data:** Current round id $t$, client set $[\mathcal{C}_r]$, aggregated Weight $\Theta_S^t$ of model with $L$ layers, support density set $d_{set}$ $= [d_1, ..., d_M]$ where $d_i < d_{i+1}$, model layer-wise parameter count $[k] = [k^1, ..., k^L]$.

**1** **if** $size(d_{set})$ *is* 1 **then**

**2** $\quad$ //JMWST subsampling in FLASH

**3** $\quad$ $\mathcal{M} \leftarrow$ `initMaskWithZeros()`

**4** $\quad$ $[\hat{d}^1, ..., \hat{d}^L] \leftarrow$ `avgLayerWiseDensity(`$[\mathcal{C}_r]$`)`

**5** $\quad$ $r_f \leftarrow \frac{d_1 \times K}{\sum_{l=1}^{L} \hat{d}^l . k^l}$

**6** $\quad$ **for** *layer* $l \leftarrow 1$ **to** $L$ **do**

**7** $\quad\quad$ `idx` $\leftarrow$ `getSortedWeightIndices(`$\Theta_S^t, l$`)`

**8** $\quad\quad$ $n_z \leftarrow$ `int(`$r_f \times \hat{d}^l \times k^l$`)` //number of non-zeros

**9** $\quad\quad$ $\mathcal{M}^l[$`idx`$[: n_z]] \leftarrow 1$

**10** $\quad$ **end**

**11** **else**

**12** $\quad$ //For hetero-FLASH

**13** $\quad$ **for** $d_i \in d_{set}$ **do**

**14** $\quad\quad$ $\mathcal{M}_i \leftarrow$ `initMaskWithZeros()`

**15** $\quad$ **end**

**16** $\quad$ $\mathcal{D}_s^t \leftarrow$ `getCurrentDensity(`$\Theta_S^t$`)`

**17** $\quad$ $[\hat{d}^1, ..., \hat{d}^L] \leftarrow$ `getLayerWiseDensity(`$\Theta_S^t$`)`

**18** $\quad$ **for** *layer* $l \leftarrow 1$ **to** $L$ **do**

**19** $\quad\quad$ `idx` $\leftarrow$ `getSortedWeightIndices(`$\Theta_S^t, l$`)`

**20** $\quad\quad$ **for** $d_i \in d_{set}$ **do**

**21** $\quad\quad\quad$ $r_{f_i} \leftarrow \frac{d_i}{\mathcal{D}_s^t}$

**22** $\quad\quad\quad$ $n_z \leftarrow$ `int(`$r_{f_i} \times \hat{d}^l \times k^l$`)`

**23** $\quad\quad\quad$ $\mathcal{M}_i{}^l[$`idx`$[: n_z]] \leftarrow 1$

**24** $\quad\quad$ **end**

**25** $\quad$ **end**

**26** **end**

---

## A.2 Model Architectures

Table 7 shows the model architectures used for MNIST and FEMNIST datasets. For CIFAR-10, CIFAR-100 and TinyImageNet we used ResNet18 (He et al., 2016) with the first `CONV` layer kernel size as $3 \times 3$ instead of original $7 \times 7$.

Table 7: Architecture used for MNIST and FEMNIST datasets

| MNIST | FEMNIST |
|---|---|
| `CONV`$5 \times 5(C_o = 10)$ | `CONV`$5 \times 5(C_o = 32)$ |
| `max_pool` | `max_pool` |
| `CONV`$5 \times 5(C_o = 20)$ | `CONV`$5 \times 5(C_o = 64)$ |
| `max_pool` | `max_pool` |
| `FC`$(5120, 50)$ | `FC`$(3136, 2048)$ |
| `FC`$(50, 10)$ | `FC`$(2028, 62)$ |

## A.3 Additional Comparisons

We now compare the performance of FLASH with that of yielded via FedSpa (Huang et al., 2022) and FedDST (Bibikar et al., 2022). For FedSpa, we implemented their proposed algorithm in our settings and kept all the hyperparameters the same for an apple-to-apple comparison. We report the best accuracy yielded for FLASH via models trained using SPDST and JMWST. As shown in Table 8, FLASH outperforms FedSpa

up to 2.41%. A similar trend is observed when we compare with FedDST, and as Table 9, on the MNIST dataset, FLASH can have an accuracy improvement of up to 1.41%.

Table 8: Comparison of FLASH with FedSpa (Huang et al., 2022) on CIFAR-10 with ResNet18.

| Data distribution | Method | Density ($d$) | Best Acc. (%) | $\delta_{Acc}$ |
|---|---|---|---|---|
| $\alpha = 1000$ | FedSpa | 0.05 | 85.63 | – |
| | FLASH | 0.05 | **87.18** | +1.55 |
| $\alpha = 0.1$ | FedSpa | 0.05 | 73.08 | – |
| | FLASH | 0.05 | **75.49** | +2.41 |

Table 9: Comparison of FLASH with FedDST (Bibikar et al., 2022) on pathologically non-IID MNIST. We used the same hyperparameter settings and models as in (Bibikar et al., 2022) for this comparison.

| Method | Density ($d$) | Communication Cost (GiB) | Best Acc. (%) | $\delta_{Acc}$ |
|---|---|---|---|---|
| FedDST | 0.2 | 1.0 | 96.10 | – |
| FLASH | | | **97.51** | +1.41 |
| FedDST | 0.2 | 2.0 | 97.35 | – |
| FLASH | | | **97.69** | +0.34 |

# B   FLASH for NLP Tasks

We also employed FLASH in fine-tuning the BERT-base (Devlin et al., 2018), a popular large language model that can be trained comprehensively with academic resources. In this experiment, the embedding layers are frozen, and since the goal is to fine-tune the model, the total number of federated rounds is 50. As shown in Table 10, the improvement in SPDST/JMWST over the baseline alternatives can also be observed on NLP tasks.

Table 10: Results for fine-tuning SST dataset on BERT-base model.

| Dataset | Data Distribution | Density ($d$) | Dense Acc % | NST Acc % | PDST Acc % | SPDST Acc % | JMWST($r_{int} = 1$) Acc % | JMWST($r_{int} = 5$) Acc % |
|---|---|---|---|---|---|---|---|---|
| SST2 | non-IID ($\alpha = 1$) | 0.2 | 92.25 | 77.59 | 78.96 | 83.07 | 80.5 | 81.78 |

# C   More Quantitative Analysis

Below, we provide more analysis and ablation to show the effectiveness of FLASH in different scenarios.

## C.1   Impact of Number of Participating Clients per Round

Fig. 11 (a) shows that JMWST and SPDST follow the same pattern at the baseline model ($d = 1.0$) with FedAvg. In other words, similar to FedAvg, as the $c_r$ increases, the performance is enhanced. Also, for a specific $c_r$, JMWST and SPDST perform better than PDST and NST.

## C.2   Impact of Batch-Normalization Layer Statistics

Fig. 11 (b) shows the performance comparison between batch normalization (BN) and static batch normalization (static BN, as suggested in (Diao et al., 2020)). In particular, in our setting, using BN layer statistics consistently outperforms the static BN.

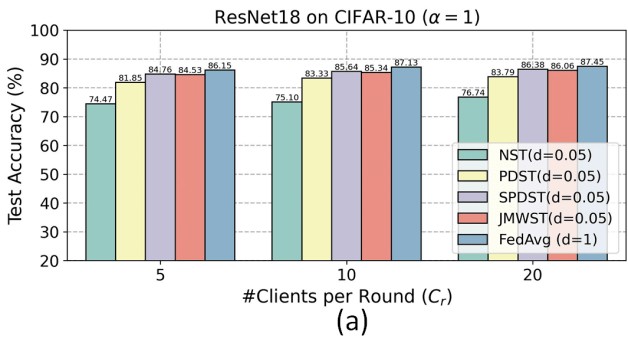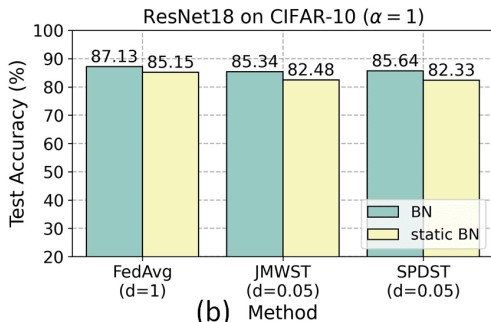

Figure 11: (a) Performance of the final trained model for different participating clients per round, (b) Significance of BN and Static BN in final model performance.

### C.3 Effect of the Mask Update Interval Rounds ($r_{int}$) in JMWST

As mentioned in the original manuscript, for JMWST, the server can increase the mask update interval ($r_{int}$) to save communication energy. We thus performed ablation for this variable from the default value of 1 (similar to (Qiu et al., 2021)) to see its impact on the final accuracy, and Table 11 and Fig. 12 show the results. In particular, as we can see in the table, less frequent update intervals can lead to better performance than the original JMWST and provide additional bidirectional saving in communication as the masks do not change every round. Fig. 12 also indicates that the improvement tends to saturate after specific $r_{int}$, which hints at the importance of this parameter. This pattern in the performance means that $r_{int}$ may potentially create a trade-off between the learnability of the masks and weights, and we believe understanding this complex trade-off is an interesting future research.

Table 11: Impact of different mask update intervals in JMWST for a target density $d = 0.1$ on CIFAR-10.

| Model | Data distribution | Mask update interval rounds ($r_{int}$) | | | |
|---|---|---|---|---|---|
| | | $r_{int} = 1$ | $r_{int} = 2$ | $r_{int} = 5$ | $r_{int} = 10$ |
| ResNet18 | IID ($\alpha = 1000$) | $87.62 \pm 0.35$ | $87.76 \pm 0.07$ | $87.86 \pm 0.13$ | $87.67 \pm 0.09$ |
| | non-IID ($\alpha = 1$) | $86.45 \pm 0.31$ | $86.26 \pm 0.07$ | $86.36 \pm 0.13$ | $86.68 \pm 0.25$ |
| | non-IID ($\alpha = 0.1$) | $74.74 \pm 1.07$ | $73.73 \pm 1.18$ | $75.47 \pm 1.08$ | $77.14 \pm 0.22$ |

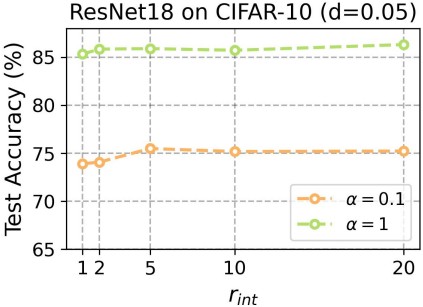

Figure 12: Test accuracy vs. mask update interval round.

### C.4 Convergence Trend of Proposed Algorithms

Fig. 13 shows the test accuracy vs. FL rounds for NST, PDST, SPDST, and JMWST algorithms on the CIFAR-10 dataset with non-IID data distribution ($\alpha = 1$). As shown in the plots, for $d = 0.05$ and $d = 0.1$, NST has slower convergence with lower final accuracy. Introducing consensus among the clients for the sparse mask accelerates the convergence and enhances the final performance.

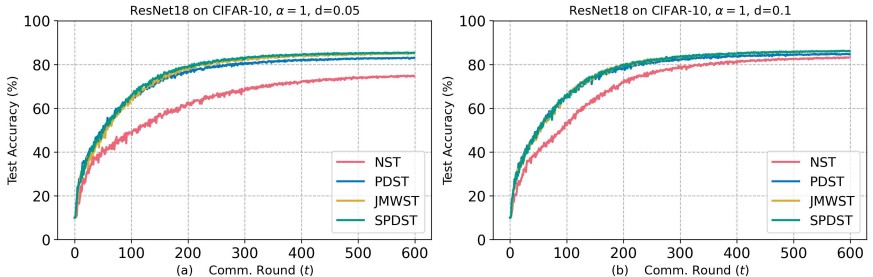

Figure 13: Performance of proposed algorithms vs. comm. rounds on CIFAR-10 dataset for (a) $d = 0.05$ (b) $d = 0.1$.

## C.5 Revisiting Sparse Mask Mismatch for NST with VGG16

Fig. 14 shows the comparison of SM between centralized and FL settings with NST on VGG16, another popular model variant. Similar to our observed trend with ResNet18, we see a significantly high SM for FL settings with a target of $d = 0.05$. This strengthens the generality of our observed limitations across different classes of DNN models.

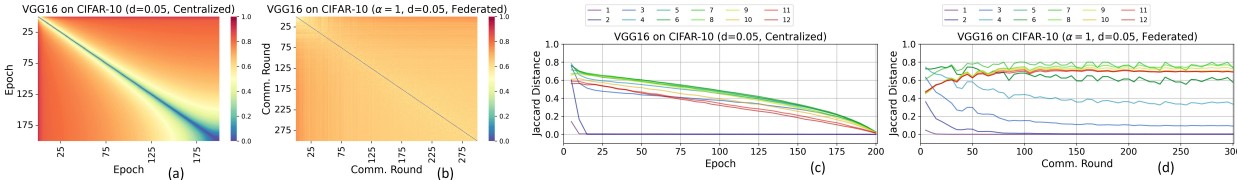

Figure 14: (a)-(b) Sparse mask mismatch (SM) for VGG16 in (a) centralized and (b) FL settings with NST. (c)-(d) Layer-wise SM vs. training epochs (rounds) for VGG16 in (c) centralized and (d) FL settings, respectively, with NST.

## C.6 Revisiting Sparse Mask Mismatch for FLASH

As demonstrated in Fig. 15, the sparse mask mismatch in the case of JMWST significantly reduces, helping the mask train in a convergent way, significantly faster than that in NST.

Fig. 16 shows the layer-wise SM for the centralized trained model (Fig. 16a) and FL trained model with sparsity (Fig. 16b-c). In particular, the SM at the later layer can significantly reduce in the case of JMWST compared to NST, further demonstrating the convergence ability even at the later layers.

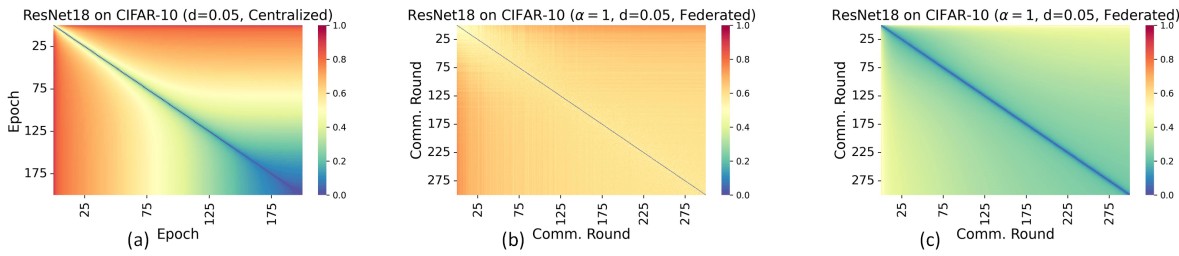

Figure 15: Sparse mask mismatch (SM) for (a) centralized sparse learning, (b) NST, and (c) JMWST in federated settings.

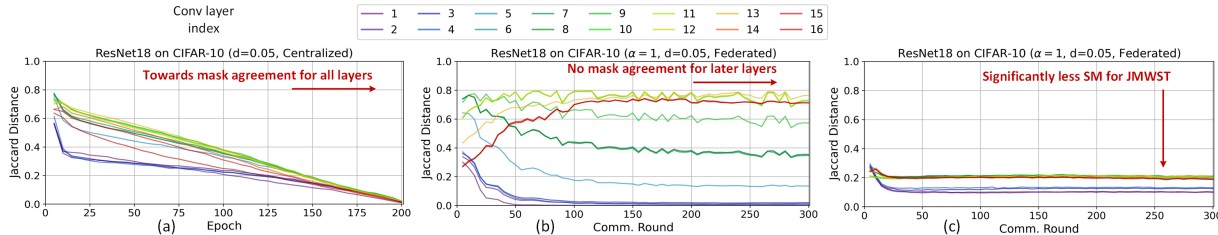

Figure 16: Layer-wise sparse mask mismatch (SM) vs. training epochs (rounds) plot for (a) centralized and (b) FL with NST, and (c) FL with JMWST.

## C.7 Sparse Mask Mismatch as a Function of d

To understand the relation of SM with $d$, we performed the baseline sparse training (NST) with ResNet18 on CIFAR-10 for three different target densities, 0.05, 0.25, 0.5. As shown in Fig. 17, the SM tends to reduce for higher density. In particular, Fig. 17(d) shows the SM for CONV layer 16 (a later layer) after round 200. The SM reduces by $1.53\times$ for $d = 0.5$ than that with $d = 0.05$, strengthening our general observation that SM becomes prominent as the density decreases.

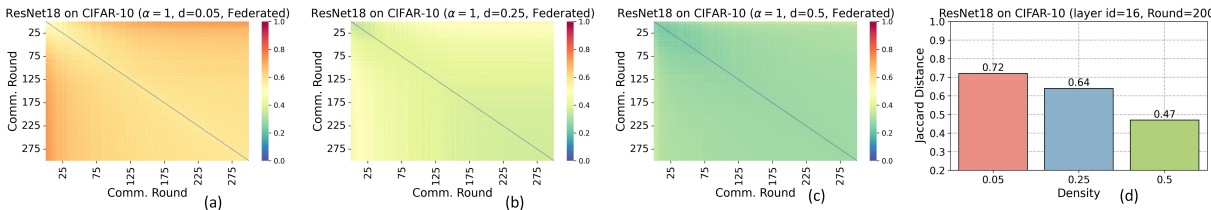

Figure 17: (a-c) SM for FL settings for three different $d$ of 0.05, 0.25, and 0.5, respectively. (d) Comparison of Jaccard distance values for the $16^{th}$ CONV layer of ResNet18 after round 200 for different $d$s.

## C.8 Sparse Mask Mismatch as a Function of Total Number of Clients

To understand the relation of SM with the number of total clients, we performed the baseline sparse training (NST) with ResNet18 on CIFAR-10 for 50 and 200 clients, respectively. As shown in Fig. 18, the SM concern persists, irrespective of the number of clients. This strengthens the generality of our observations over the total number of clients.

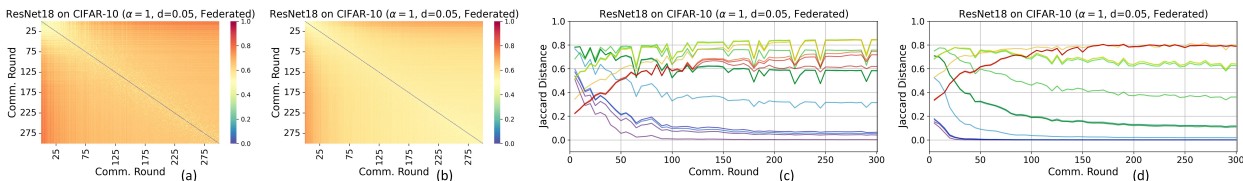

Figure 18: (a-b) SM for FL settings for (a) 50 and (b) 200 clients. (c-d) Layer-wise SM vs. training rounds for (c) 50 and (d) 200 clients.

## D Model Personalization vs. Mask Consensus

Compared to personalized FL, FLASH comes with a different objective. Our goal is to train a single global model that performs well on all clients' datasets. Mask convergence plays a key role in maintaining stable

learning progress and high model accuracy, especially for heterogeneous data and high compression ratios. However, model personalization aims to find individual models that adapt to each client.

# E  Discussion on Computing Benefits of Sparse Learning at the Edge

To extract FLOPs benefits for irregular pruning in FLASH, we assume that the compute energy for the sparse network can be avoided via the means of clock-gating (Yang & Kim, 2018) of the zero-valued weights. Moreover, there has been recent development of sparsity-friendly DNN accelerators (Qin et al., 2020) that can efficiently reduce the compute cost by a significant margin. Such accelerators can leverage the yielded sparse FL models to deploy at compute-constrained edges.

## E.1  FLOPs vs. Communication Cost for Different Density Budgets

To reach a target accuracy value, we plot the FLOPs to uplink communication cost for different density budgets in Fig. 19.

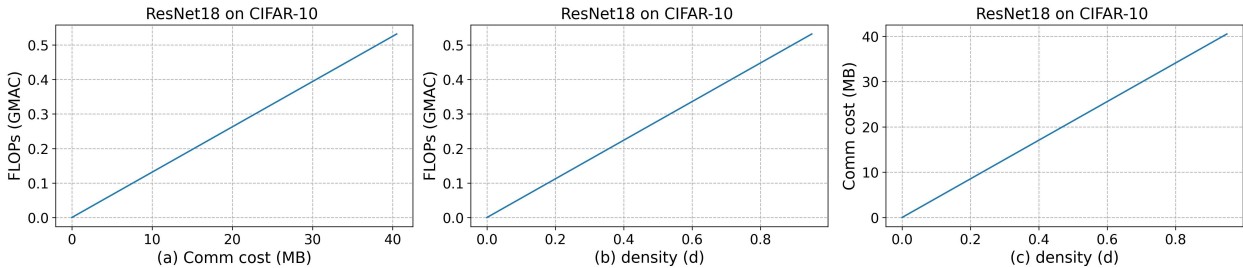

Figure 19: Computation and Communication relation with (a) each other (b, c) with different density levels for SPDST algorithm.

## E.2  Clarification on Sparse Training vs. Sparse Updates

Here, we want to emphasize that our framework differs from the method proposed for sparsifying updates such as top-K. In particular, in algorithms that try to sparsify the updates, the clients must do a dense gradient update, costing higher computation and potential communication overhead. Our framework, specifically SPDST, on the contrary, ensures only k weights are updated to be non-zero during each round for each client, allowing us to yield the lucrative benefits of sparse gradient computation. Moreover, it ensures each client sends and receives only k weights to and from the server at the end and beginning of each round. Finally, in every round of the top-k algorithm, clients ignore the weights not in the top-k, potentially causing *wasted* computation and performance degradation. However, in SPDST, such *wasted* computations are avoided since only a fixed fraction of weights are trained.

## E.3  Computation Saving in FLASH

Employing sparse learning in FL helps participating clients reduce communication and compute costs (FLOPs) for training. Without the loss of generality, we now evaluate the convolutional layer training FLOPs for FLASH and demonstrate the relation of parameter density $d$ with the reduction in FLOPs and communication cost.

The training FLOPs for a layer $l$ ($F_{layer}^l$) can be partitioned into forward operation FLOPs ($F_{fwd}^l$), backward input ($F_{back\_in}^l$) and weight gradient ($F_{back\_wt}^l$) compute FLOPs. With the assumption of the no-compute cost associated with the zero-valued weights via zero-gating logic (Kundu et al., 2023b), the $F_{layer}^l$ for FLASH with parameter density $d$ ($d << 1.0$) is

$$F_{layer}^l = d \times [Fu_{fwd}^l + Fu_{back\_in}^l] + s_a \times Fu_{back\_wt}^l \tag{4}$$

If the zero weights' gradients flow is computed for mask learning, then $F_{back\_wt}$ can't leverage the advantage of low parameter density. Thus, gradients are dense in JMWST, and $F_{back\_wt}$ is the same as that in dense computation. However, for SPDST or JMWST with $r_{int} > 1$, zero weights remain zero, allowing us to skip the associated gradient computation safely. This helps extract the benefits of sparsity during all three stages of FLOPs computation. In other words, FLASH can improve communication and compute costs for clients with limited resources.

### E.4    Evaluation of the Communication Cost

Communication cost is associated with transmitting the newly updated weight (or gradients) from client to server or vice versa. The communication reduction is achieved by making the weight matrices sparse. In this way, clients do not need to send the whole matrix; instead, they can only send the value of the non-zero ones. In our experiments, we use the compressed sparse row (CSR) format of the sparse model weights to communicate them between clients and the server. Finally, communication-saving is evaluated by the ratio of full dense model communication cost to sparse model in CSR format.

### E.5    Support for Hardware-Friendly Sparsity Patterns

Irregular sparsity is often not well-suited for hardware benefits without dedicated architecture or compiler support. However, we can yield computation energy saving with custom zero-gating logic (Kundu et al., 2020) and compiler support (Liu et al., 2018). As we intended to achieve reduced computation energy and communication cost in FL and as irregular sparsity can yield higher compression than structured/pattern sparsity, we have limited our evaluations to random or irregular sparsity only. Nevertheless, we believe our framework can support more complex and structured pruning, which we briefly explain.

Among the various hardware-friendly sparsity patterns, the recently proposed $N : M$ sparsity (Zhou et al., 2021) has gained significant attention due to its less strict constraints. For SPDST, post `Stage 1`, sparse mask selection can be easily extended to support the $N : M$ sparsity. In particular, for a layer $l$, instead of random assignment of $d^l \times k^l$ non-zero mask locations, we can partition the total non-zero elements into $G^l$ groups, where each group will contain $d^l \times k^l/G^l$ non-zero elements. Here, $G^l$ is evaluated as $k^l/M$, $M$ representing the total element size out of which we need to have a certain fraction as non-zero, and $k^l$ represents the total number of weights for that layer. As the masks remain frozen, we maintain the pattern throughout the training for each client to extract the benefit. For JMWST, we can adapt this principle in the prune and regrow policy during each client's local training. In Table 12, we now show results with pattern pruning with $N : M$ sparsity for the CIFAR-10 dataset.

Table 12: FLASH with structured sparsity for CIFAR-10 dataset

| Data Distribution | Density | Method | Acc (%) |
|---|---|---|---|
| non-IID ($\alpha = 1$) | 0.2 | SPDST | 86.96 |
| non-IID($\alpha = 0.1$) | 0.2 | SPDST | 77.64 |

