# OpenReview forum: "Revisiting Sparsity Hunting in Federated Learning: Why does Sparsity Consensus Matter?"
_TMLR — Accepted by TMLR_

### Review · Reviewer_WFQs · 2023-07-29

**Summary Of Contributions:**

This paper studies sparse federated learning, particularly for resource-limited computing devices. The paper observed that directly adopting sparse learning algorithms can cause the lack of consensus of the learned masks, which will further cause the degradation of the learned global model. Based on the observation, the paper proposed the federated lottery aware sparsity hunting (FLASH) framework to achieve both computation and communication efficiency in FL using sparse learning. Experimental results have been shown to demonstrate the effectiveness of the proposed FLASH framework.

**Audience:**

Yes

**Broader Impact Concerns:**

I don't find any concerns on the ethical implications of the work.

**Claims And Evidence:**

Yes

**Requested Changes:**

- Elaborating more on the actual computation efficiency gain of the proposed FLASH framework.
- Adding larger-scale computer vision (e.g., ImageNet) and natural language processing (e.g., GPT finetuning) experiments.
- Provide more guidance on selecting between SPDST and JMWST in practice.

**Strengths And Weaknesses:**

Strengths:
- The paper is generally well-written. Improving the computation and communication efficiency in federated learning is of potential importance.
- The idea behind the proposed FLASH framework is easy to follow.
- The experimental results of the FLASH framework (at least for computer vision tasks) are convincing.

Weaknesses
- The paper keeps claiming "computation efficiency". However, I think FLASH can only introduce random sparsity on local models. For the current generation of hardware, sparse computing is not well supported. Thus, with respect to the actual compute speed, I am not sure if FLASH can attain any speedup.
- All experiments are on small-scale vision tasks. It would be great to explore language modeling tasks on larger-scale datasets and models.
- Two approaches have been proposed in FLASH (i.e., SPDST and JMWST), however, it is not clear which approach users should use in practice.

---

> ### Author Response · Authors · 2023-08-15
> **Response to reviewer WFQs**
>
> We thank the reviewer for their constructive comments and praise for the work. We now address the concerns raised by the reviewer.
>
>     On actual speed-up
> The reviewer has highlighted an important point here. It is true that with irregular sparse weight tensors, it may not often be possible to have significant speedup. However, recently there have been significant efforts in computer architecture research  [1, 2] to speed up such irregular computations with additional compiler, architecture, or package support. We thus believe with such advancement, it is possible to leverage the benefit of speeding up, particularly for the scenario of ultra-high compression that we have considered here.
>
> We have provided details on the computation gains and comparison of SPDST with JMWST in terms of compute benefit during training in appendix section B. under the title: **Computation saving for FLASH**. We will be happy to expand on this in case the reviewer has any further questions.
>
>     Adding larger-scale computer vision (e.g., ImageNet) and natural language processing (e.g., GPT finetuning) experiments.
>  We appreciate the reviewer’s suggestion on a large-scale dataset. We now provide results with the mini-ImageNet dataset with a high image resolution size 224x224x3 and demonstrate the efficacy of our methods in Table R3T1.
> Mini-ImageNet on ResNet 34, density = 0.05, alpha=1, Federated Round= 600
>
> #### R3T1
> Method | Dense | NST | PDST | SDPST | JMWST ($r_{int} = 1$) | JMWST ($r_{int} = 5$)
> |---|---|---|---|---|---|---|
> Accuracy | 49.88 | 36.2 | 46.54 | 48.16 | 46.84 | 47.48
> -----
> On GPT type models: We appreciate the reviewer’s feedback. Current LLMs often demand significant pre-training, model storage, and data (e.g., LLaMA [4] open-sourced by Meta costs approximately $5M USD to train with 1 Trillion tokens), making them difficult to study in academic environments. Thus the extension to LLMs is out of the scope of the current manuscript, however, is an important future research direction. We will add this to our conclusions.
> Having said this, we now demonstrate results for fine-tuning the BERT-base [3], a popular large language model that can be trained comprehensively with academic resources. As shown in R3T2, the improvement in SPDST/JMWST over the baseline alternatives can be observed on NLP tasks as well.
>
> #### R3T2
> SST2 dataset on Bert Model (density=0.2, alpha=1, freeze embedding layers, federated rounds = 50)
> |Method | Dense | NST | PDST | SDPST | JMWST ($r_{int} = 1$) | JMWST ($r_{int} = 5$)|
> |---|---|---|---|---|---|---|
> |Accuracy | 92.25 |77.59| 78.96 | 83.07 | 80.5| 81.78|
>
>     On guidance for selection between SPDST and JMWST
> The main difference between SPDST and JMWST is that the sparsity mask remains fixed for the former. As a result, if the setting is more stable and clients in different rounds have more similar data distribution, using SPDST helps to focus on the training weight only. This further ensures both forward and backward compute and communication benefits, proportional to the compression benefits. On the other hand, JMWST allows the change of sparsity masks after a certain period of rounds. Ideally, if the clients have heterogeneous datasets, training the mask can improve performance, particularly during the early stages of the training rounds. It is notable that if we set $r_{int}=1$, then the pruned weights do not get enough fine-tuned. Therefore, we believe there is a tradeoff between changing the mask and fine-tuning the weights. This tradeoff can be controlled with $r_{int}$, and as presented in Table 3, for the majority of the heterogeneous data distributions, $r_{int}=5$ gives the highest accuracy.  Further insight into understanding the optimal update interval and number of rounds for better convergence with JMWST is left for future research.
>
> [1] Qin, Eric, et al. "Sigma: A sparse and irregular gemm accelerator with flexible interconnects for dnn training." 2020 IEEE International Symposium on High Performance Computer Architecture (HPCA). IEEE, 2020.
>
> [2] Li, Zhiyao, et al. "Spada: Accelerating Sparse Matrix Multiplication with Adaptive Dataflow." Proceedings of the 28th ACM International Conference on Architectural Support for Programming Languages and Operating Systems, Volume 2. 2023.
>
> [3]  Devlin, Jacob, et al. "Bert: Pre-training of deep bidirectional transformers for language understanding." ACL 2019.
>
>
> [4] Hugo Touvron et al. LLaMA: Open and efficient foundation language models. arXiv:2302.13971, 2023

---

### Review · Reviewer_5Y9b · 2023-08-04

**Summary Of Contributions:**

This paper focusses on the problem of developing sparse learning algorithms for federated learning (FL). Based on some observation, the authors conjecture that the failure of conventional naive sparse training (NST) method is originated from the lack of consensus in the trained sparsity masks among the clients.

The authors propose FLASH, a new two-staged framework for applying sparse learning on FL. Once the masks are initialized in the first stage, FLASH apply sensitivity-aware training methods (dubbed as SPDST and JMWST).

Experimental results show that the proposed methods successfully do sparse learning, compared with existing methods.

**Audience:**

Yes

**Claims And Evidence:**

Yes

**Requested Changes:**

- Add theoretical interpretation suggested above
- Fix the writing issue listed above
- Are all the experiments focus on specific local sparse learning (SL) method? If so, adding more experimental results on various local SL methods would be beneficial


**Strengths And Weaknesses:**

**Strengths**
- The empirical result is interesting, especially the success of SPDST which freezes the mask and do client model updating only for the survived links.

**Weaknesses**
- (Lack of theoretical interpretation) I think it would be great if the authors provide theoretical explanation on the success of the proposed methods, at least for very simple architecture & data setting. Probably focusing on some toy example that NST fails, but FLASH successes is enough.

- (Room for improvement on writing) I think this paper has much room for improvement on writing. I want the authors to clarify the following questions I have
    - Fig.4 shows Jaccard distance between two models? Which two models are chosen? Global model at round $i$ and round $i+1$?
    - While explaining Table 2, I could find some weird phrases, e.g., “for the training in row 1”. The terminology “pre-defined w/ mask frozen” is also weird. What is pre-defined?
        - I couldn’t fully get the schemes in Table 2. Especially, what is the scheme for 2nd row? Do we define the mask based on sensitivity, and then train the survived weights while the mask is frozen?
        - The author says “pruning sensitivity” is evaluated using another pre-trained model, but how is it trained? I think it is better to provide some visualization of schemes considered in Table 2.
    - In Table 2 (from row 3 to 5), how do we choose a model that meet the target SM value? We stop training when the average Jaccard distance Fig.4b is less than the target?

---

> ### Author Response · Authors · 2023-08-16
> **Response to reviewer 5Y9b part (1/2)**
>
> We thank the reviewer for their constructive comments and praise for the work. We now address the concerns raised by the reviewer.
>
>     Theoretical interpretation of the observation
> In our view, one important way to understand various machine learning phenomena as they occur in practice is to investigate them empirically as they occur in practice. The distinctive quality of our observations is the level of depth and rigor with which we analyzed the differences in sparse learning between centralized and FL settings in terms of **novel empirical demonstrations**. We further utilized these observations to present a *computation and communication efficient* sparse learning particularly tailored for FL. Finally, our extensive demonstrations of various datasets with different models validate the importance of our observations in closing the gap in the accuracy of a sparsely trained model from that of an unpruned baseline. While theoretical understanding is important, we believe such thorough empirical demonstration at scale has its own merit. The recent acceptance of ZeroFL [1] in ICLR 2022 shows the importance of such empirical contributions that have been acknowledged by the top-venue reviewing committee. We would be happy to take up the theoretical foundation of these observations as a future research direction.
>
> Towards that, we believe our work has empirically established an intriguing relation between sparse mask mismatch and server model accuracy for a given total round of training. To strengthen the observation, we have performed extensive evaluations of the proposed methods over both large-scale image classification and  NLP datasets across CNNs and transformers. We thus hope this will inspire the research community to delve further into this interesting observation.
>
> Additionally, we followed the suggestion of the reviewer and provided a theoretical discussion on the convergence of the proposed sparse learning in section C of the Appendix of the updated manuscript. Besides, the convergence of the FedAvg is controlled by the variance of the stochastic gradient at each client. However, we believe that masking the gradient would potentially decrease this variance and hence improve the convergence. We will be happy to incorporate this as a part of the future work.
>
> Intuitively, the naive sparse training does not necessarily restrict a common non-zero starting point of the weight for all the clients. This may, in turn, add noise to the updated gradient both in terms of averaging via participating clients (meaning averaging a weight corresponding to the ones for which it is non-zero). Additionally, allowing sparse updates for each client may also introduce mask noise due to spurious updates because of some training samples. A similar observation is made by FjORD [2] while performing ordered pruning. Allowing all clients to have consistent masks avoids both the source of gradient averaging noise as well as mask update noise. Having said this, we believe further investigation on empirical observation is interesting for future research.
>
>     On Improvement in Writing
> We apologize for the lack of clarity in writing. The draft is now updated to address the mentioned concerns.
>
> **Jaccard distance in Fig. 4**. We would like to clarify that this Jaccard distance is measured for the same model (global model) between its sparse masks at the end of rounds *i* and *i+5*.
>
> **pre-defined with mask frozen**. We apologize for the confusion. We would like to highlight that a “pre-defined mask” is a mask that is determined before training, usually in a data-independent way. This definition is in line with [3], one of the works that introduced pre-defined sparsity for convolution layers. “Frozen” highlights the fact that the sparsity mask does not change and remains fixed throughout the training process. We will be glad to simplify these terminologies for the broader community to understand.
>
> **scheme for 2nd row in Table 2**. We confirm that the reviewer’s understanding of 2nd row is correct. We have mentioned this in the “Use sensitivity” column of Table 2. However, for row 1, we create the mask with equal sensitivity in each layer, as now also clarified in the writing.
>
> **pruning sensitivity evaluation on another model**. We first train an iso architecture model to meet the target density $d$. Then, for all layers, we measure the pruning sensitivity following Eq. 2 of the manuscript. Then, we use this sensitivity to predefine a "sensitivity-driven" mask in the experiments of rows 2-5 for a model with the same architecture but initialized from scratch.
>
> please find the rest of our response in Part 2/2

---

> > ### Author Response · Authors · 2023-08-16
> > **Response to reviewer 5Y9b part (2/2)**
> >
> > On Improvement in Writing (contd.)
> >
> > **choose a model with the target SM**. We apologize for the confusion here. For the experiments in rows 3-5, we forcefully update the sparsity mask at the end of each epoch such that the SM between epoch i and i+1 maintain the value mentioned in the corresponding table. However, while updating the mask, we also ensure the sensitivity of the layers is still maintained. To clarify this, we added this sentence: “For rows 3-5 at each epoch, a fraction of the mask in the specified layers changes to meet the target SM value”.
> >
> >     Specific local sparse learning method
> > In our paper, we used various popular methods of local sparse training, including [8], erk/erk+ (initialization-driven sparse learning) [7], pre-defined sparser learning [3], SL method used by ZeroFL[1], namely SWAT[5]. However, similar to other sota FL sparse learning (ZeroFL[1], FedSpa[4], FedDST[6]) majority of these suffer from non-negligible accuracy drop at ultra-high sparsity. We showed a comparison with alternate methods of sparse learning in Fig. 10 of the main manuscript.
> >
> > [1] X. Qiu, et al. "ZeroFL: Efficient on-device training for federated learning with local sparsity.", ICLR 2022
> >
> > [2] S. Horvath, et al. "Fjord: Fair and accurate federated learning under heterogeneous targets with ordered dropout."NeurIPS 2021.
> >
> > [3] S. Kundu et al., “Pre-defined Sparsity for Low Complexity Convolutional Neural Networks”, IEEE Transactions on Computer, 2020.
> >
> > [4] T. Huang, et al. "Achieving personalized federated learning with sparse local models." arXiv preprint arXiv:2201.11380 (2022).
> >
> > [5] M. Raihan, et al. "Sparse weight activation training." Advances in Neural Information Processing Systems 33 (2020): 15625-15638.
> >
> > [6] S. Bibikar, et al. "Federated dynamic sparse training: Computing less, communicating less, yet learning better." AAAI, 2022.
> >
> > [7] U. Evci, et al. "Rigging the lottery: Making all tickets winners." ICML, 2020.
> >
> > [8] T. Dettmers. Sparse networks from scratch: Faster training without losing performance. arXiv preprint arXiv:1907.04840, 2019.

---

### Review · Reviewer_W8Q4 · 2023-08-08

**Summary Of Contributions:**

This paper proposes to improve federated learning sparse training method by introducing sparse training mask consensus to enhance the final accuracy. The authors first observe that the closer the sparse training masks are between training epochs, the better the model converges. Based on this observation, they propose two methods: SPDST and JMWST. SPDST: The server defines a mask before the training starts and keeps it unchanged throughout the training process. JMWST: Each client adjusts their mask locally, and the server uses the aggregated model to sample the new mask, then broadcast it to the clients. Experimental results demonstrate that a high mask consensus is beneficial in model convergence.

**Audience:**

Yes

**Broader Impact Concerns:**

N/A.

**Claims And Evidence:**

Yes

**Requested Changes:**

The above three weaknesses are critical to the work.

**Strengths And Weaknesses:**

Strengths

1.	The author conducted numerous experiments and made several observations, which make their claim about mask consensus promising. The authors proposed a FLASH method that can achieve both computation and communication efficiency in FL by employing sparse learning.

2.	The author presents a straightforward takeaway: the smaller the variation of sparse masks during the training process, the better the performance.

3.	The extensive evaluation on real image datasets demonstrates the effectiveness of the proposed approach for sparse learning in federated settings.


Weaknesses

1.	This paper presents interesting new findings, but its technical novelty is incremental. All technique aspects, including mask initialization, SPDST, JMWST, and hetero-FLASH are either based on existing techniques or making minor adjustments.

2.	This paper needs to provide a theoretical analysis of the critical observation mentioned in Strengths 2, which lowers the generalizability and reliability of this observation.

3.	The paper may also consider discussing the impact of data heterogeneity and mask consensus. Many previous works suggested that data heterogeneity can result in different importance and mask consensus among clients, i.e., personalization. However, the results of this paper indicate that mask consensus should be maintained even under high non-IID degrees.

---

> ### Author Response · Authors · 2023-08-15
> **Response to reviewer W8Q4 part (1/2)**
>
> We thank the reviewer for their constructive comments and praise for the work. We now address the concerns raised by the reviewer.
>
>     Incremental technical novelty
>
> While some sparse learning techniques have been exploited in centralized settings, sparse learning in federated settings poses new challenges. FLASH is aimed at attacking these new challenges. We iterate our important contributions as follows:
>
> **Investigating a more effective pruning sensitivity metric in FL settings.** We demonstrate that the sensitivity metric (stage 1) in the paper is more effective in FL settings compared to existing approaches like ERK or ERK+ [1]. The sensitive metric only incurs negligible computation and communication overhead.
>
> **Key observation on the importance of mask agreement in FL settings.** Different from prior works, FLASH is the first work revealing that ensuring agreement of mask in sparse FL is crucial to main training performance, especially under high compression.
> We demonstrate that one-shot mask assignment, based on the sensitivity evaluated at stage 1, results in higher accuracy in FL settings at ultra-high compression. And such a finding applies to both i.i.d and non-i.i.d data distributions.  We believe the revealing of mask agreement in FL settings has profound impacts on sparse FL.
>
> **Sparse learning for heterogeneous devices.** We are the first to present an FL pruning scheme for devices with different resource budgets and demonstrate the efficacy of our method in that practical scenario.
>
> **Extensive empirical validation.** We believe that different evaluations on various datasets like Tiny-ImageNet, (and now ImageNet in Table R3T1) and on NLP datasets using a large language model like BERT-base (table R3T2) add to the contribution of our work. No prior works demonstrate the efficacy of sparse learning on various settings and datasets for both computer vision and NLP.
> Thus we would like to respectfully disagree with the reviewer as we believe that although some aspects of the solution may be simple, the systematic pathway and observations to reach those solutions are important contributions.
>
> **Impact**. Prior SOTA works such as ZeroFL[3], FedDST[4], and FedSPA [2] achieved computation efficiency at the cost of a large performance gap compared to dense model training. Moreover, they do not have communication benefits proportional to compression ratios. In our paper, we first dig into potential reasons for poor performance in FL as opposed to centralized sparse learning, then present FLASH to mitigate the gap and empirically demonstrate that FLASH yields new SOTA in sparse FL.
>
>     On Theoretical analysis.
> In our view, one important way to understand various machine learning phenomena as they occur in practice is to investigate them empirically as they occur in practice. The distinctive quality of our observations is the level of depth and rigor with which we analyzed the differences in sparse learning between centralized and FL settings in terms of **novel empirical demonstrations**. We further utilized these observations to present a *computation and communication efficient* sparse learning particularly tailored for FL. Finally, our extensive demonstrations of various datasets with different models validate the importance of our observations in closing the gap in the accuracy of a sparsely trained model from that of an unpruned baseline. While theoretical understanding is important, we believe such thorough empirical demonstration at scale has its own merit. The recent acceptance of ZeroFL [3] in ICLR 2022 shows the importance of such empirical contributions that have been acknowledged by the top-venue reviewing committee. We would be happy to take up the theoretical foundation of these observations as a future research direction.
>
> Towards that, we believe our work has empirically established an intriguing relation between sparse mask mismatch and server model accuracy for a given total round of training. To strengthen the observation, we have performed extensive evaluations of the proposed methods over both large-scale image classification and  NLP datasets across CNNs and transformers. We thus hope this will inspire the research community to delve further into this interesting observation.
>
> Additionally, we followed the suggestion of the reviewer and provided a theoretical discussion on the convergence of the proposed sparse learning in section C of the Appendix of the updated manuscript. Besides, the convergence of the FedAvg is controlled by the variance of the stochastic gradient at each client. However, we believe that masking the gradient would potentially decrease this variance and hence improve the convergence. We will be happy to incorporate this as a part of the future work.
>
> please find the rest of our response in Part 2/2

---

> > ### Author Response · Authors · 2023-08-15
> > **Response to reviewer W8Q4 part (2/2)**
> >
> >     On theoretical analysis (contd.)
> >
> > Intuitively, the naive sparse training does not necessarily restrict a common non-zero starting point of the weight for all the clients.  This may, in turn, add noise to the updated gradient both in terms of averaging via participating clients (meaning averaging a weight corresponding to the ones for which it is non-zero). Additionally, allowing sparse updates for each client may also introduce mask noise due to spurious updates because of some training samples.  A similar observation is made by FjORD [5] while performing ordered pruning. Allowing all clients to have consistent masks avoids both the source of gradient averaging noise as well as mask update noise. Having said this, we believe further investigation on empirical observation is interesting for future research.
> >
> >     On model personalization with heterogeneous mask and data heterogeneity
> > Compared to personalized FL, FLASH comes with a different objective. Our goal is to train a single global model that performs well on all clients' datasets. Mask convergence plays a key role in maintaining stable learning progress and high model accuracy, especially for heterogeneous data and high compression ratios. However, model personalization is aimed at finding individual models that adapt to each client.
> >
> > [1] U. Evci, et al. "Rigging the lottery: Making all tickets winners." In ICML, pp. 2943-2952. PMLR, 2020.
> >
> > [2] T. Huang, et al, and Dacheng Tao. "Achieving personalized federated learning with sparse local models." arXiv preprint arXiv:2201.11380 (2022).
> >
> > [3] X. Qiu, et al. "ZeroFL: Efficient on-device training for federated learning with local sparsity.", ICLR 2022
> >
> > [4] S. Bibikar, et al. "Federated dynamic sparse training: Computing less, communicating less, yet learning better." AAAI 2022.
> >
> > [5] S. Horvath, et al. "Fjord: Fair and accurate federated learning under heterogeneous targets with ordered dropout."NeurIPS 2021.

---

### Decision · Action_Editors · 2023-09-11

**Recommendation:** Accept with minor revision

**Comment:**

The paper provides clear evidence that (1) lack of consensus among sparsity masks in federated learning settings harms convergence an (2) the proposed sparsity masking methods offer clear benefits over related methods in the field. The reviewers agreed that the experiments were quite convincing and thorough in the computer vision setting, and the added experiments in NLP tasks only heighten this.

## Requested changes

Some of the reviewers were concerned with theoretical interpretations of the behavior in this work. I believe that this is unnecessary to provide (TMLR guidelines do not suggest that any paper must have experiments and theory). While I am impressed by the speed with which the authors added a theoretical analysis to the paper, **I would like to request that the authors remove this from the final version**. I believe that this theoretical analysis has not been subject to appropriate peer review, and that it is frankly unnecessary, and possibly a distraction from the strong empirical evidence in the paper.

Reviewers also requested changes related to the organization, clarity, and writing. I would urge the authors to do one last significant pass in improving these, though I wish to emphasize that the authors have **already done a great job of improving the readability of the paper**. I would also urge the authors to ensure that their citation formatting is correct throughout the paper. There is still at least one place where the authors use `\cite` instead of `\citet` or `citep` (the conclusion has a line "compared to SOTA Qiu et al., (2021)" which is difficult to parse).

**Audience:**

Yes. There is a large amount of work on federated learning, and communication-efficiency is an important topic in the field. There are clearly people who are working on related (or even the same) problem, and the paper may be of interest more generally to audiences interested in sparsity. In particular, the notion of consensus in sparsity masks governing speed of convergence is one that I believe some of TMLR's audience would be interested in.

**Claims And Evidence:**

The reviewers have all concurred that the experimental evidence for the FLASH framework proposed by the authors are strong. The authors provide substantial evidence that FLASH can improve computation- and communication-efficiency gains in federated learning. Moreover, the authors convincingly showcase the notion that the "lack of consensus" among sparsity masks can slow down convergence.

I will note for posterity that multiple reviewers asked for theory that explains the empirical success of the authors' methods. I believe this is unwarranted, as it does not match the TMLR reviewing style. While the authors did supply a theoretical analysis in a post-discussion paper revision, I do not believe that this is necessary in this regard. The empirical evidence is sufficient to demonstrate the claims of the authors.

---

> ### Author Response · Authors · 2023-09-25
> **Thank the AE and upload camera-ready version**
>
> Dear Action Editor,
>
> We want to express our sincere gratitude for your time, efforts, and attention during the review process and thank you for your feedback and insights on our paper.
>
> We have uploaded the camera-ready version. In this version, we have removed the theoretical results from the appendix, improved the writing, and added more clarifications. Furthermore, we have shared our codebase.
>
> We would genuinely appreciate it if you could inform us whether there are any remaining concerns.
>
> With Regards,
>
> Authors